# POSITIONAL ENCODING VIA TOKEN-AWARE PHASE ATTENTION

## ABSTRACT

We prove under practical assumptions that Rotary Positional Embedding (RoPE) introduces an intrinsic distance-dependent bias in attention scores that limits RoPE's ability to model long-context. RoPE extension methods may alleviate this issue, but they typically require post-hoc adjustments after pretraining, such as rescaling or hyperparameters retuning. This paper introduces Token-Aware Phase Attention (TAPA), a new positional encoding method that incorporates a learnable phase function into the attention mechanism. TAPA preserves token interactions over long range, extends to longer contexts with direct and light fine-tuning, extrapolates to unseen lengths, and attains significantly lower perplexity on long-context than RoPE families.

## 1 INTRODUCTION

Rotary Positional Embedding (RoPE) (Su et al., 2021) is a widely adopted positional encoding method in transformers (Vaswani et al., 2017) that applies complex rotations to token representations.

However, RoPE, as originally designed, is not able to extrapolate to context lengths that were not seen during pretraining (Sun et al., 2022), even with extensive continual pretraining at the extended lengths (Chen et al., 2023; Xiong et al., 2023). Various extension methods are proposed to improve RoPE's ability to adapt to longer context, such as increasing RoPE base frequency (Xiong et al., 2023), Position-Interpolation (Chen et al., 2023; Peng et al., 2023b; Ding et al., 2024), YaRN (Peng et al., 2023c) etc.

Many popular publicly available open-source large language models (LLMs) adopt RoPE as their default positional encoding strategy and apply certain RoPE extension methods after pretraining, including LLaMA (Touvron et al., 2023; Dubey et al., 2024), DeepSeek (Team, 2024), Qwen (Bai et al., 2024), Mistral (Jiang et al., 2023), Phi (Abdin et al., 2024a;b), Kimi (Team et al., 2025), PaLM (Chowdhery et al., 2022) etc.

Despite RoPE's widespread use in modern LLMs, the reasons behind RoPE's limitations and extensions remain poorly understood. This theory gap motivates our study. In this paper, we prove that RoPE attention carries a non-trivial distance bias—that is, the attention magnitude is dominated by distance between token positions rather than content. This is apparently undesired in language modeling, because relevant information may be downplayed just because it's in a "bad" position. In addition, our proof shows that certain RoPE extension methods such as reducing base-frequency and PI indeed mitigate this bias issue.

While RoPE extensions help, they remain tied to the rotary structure, and rely on manual interventions after pretraining, such as applying an ad hoc formula to rescale input positions, or adjusting base frequency through extensive empirical tuning. The need for such unnatural post-hoc modifications suggests that a more fundamental limitation is present in RoPE's design. Because ideally, a robust positional encoding scheme should be able to fit longer context with minimal fine-tuning and, more importantly, without either hyperparameter or input changes.

We introduce Token-Aware Phase Attention (TAPA), a simple positional encoding framework that inserts a learnable phase function into the attention mechanism. TAPA eliminates undesired distance bias and preserves interactions

with long-range context. Importantly, it extends a pretrained model to longer contexts via a direct fine-tuning, without input tweaks or hyperparameter retuning.

Empirically, we pretrain a transformer with LLaMA3 7B model architecture at 8k, fine-tune at 32k, and evaluate up to 64k (Table 1). TAPA matches baselines through 16k. At 32k, TAPA reaches 11.74 perplexity, reducing perplexity by $\sim$9.4% vs. RoPE/PI and $\sim$3.5% vs. YaRN. At 64k, TAPA remains $\sim$11.75 while others blow up to $\sim$2$\times$10$^3$, making it the only method whose test perplexity continues decaying up to 49k and remain non-collapsing at 64k.

## 2 THEORETICAL ESTIMATES FOR ROTARY POSITIONAL EMBEDDING (ROPE)

### 2.1 BACKGROUND

We recall the details of RoPE (Su et al., 2021), and introduce the notations that are important for our future analysis.

**Notation 2.1** (RoPE). We let $D$ be transformer head dimension, $1/\theta_0$ be RoPE base frequency, and $\theta_d = \theta_0^{2d/D}$ be the rotation argument of the $d$-th dimension [1].

Denote by $q^{(m)}$ and $k^{(n)}$ the query and key vector representations for tokens at position $m$ and $n$. When no ambiguity is present, we shall drop the upper indices $m$ and $n$ to simplify notations. Denote by $q_{[2d:2d+1]}, k_{[2d:2d+1]} \in \mathbb{R}^2$ the 2-dimensional real vectors that consist of the $(2d)^{\text{th}}$ and $(2d+1)^{\text{th}}$ coordinates of $q$ and $k$ (for $0 \le d \le D/2 - 1$). Further, we complexify both vectors into $q_{[2d:2d+1]}^{\mathbb{C}}$ and $k_{[2d:2d+1]}^{\mathbb{C}}$; that is,

$$
\begin{aligned}
q_{[2d:2d+1]}^{\mathbb{C}} &= q_{2d} + i \cdot q_{2d+1}, \\
k_{[2d:2d+1]}^{\mathbb{C}} &= k_{2d} + i \cdot k_{2d+1}.
\end{aligned}
\tag{1}
$$

The RoPE attention score between $q$ at position $m$ and $k$ at position $n$ is defined by

$$
\text{Attn}_{\text{RoPE}}(q,k) = \frac{1}{\sqrt{D}} \text{Re}\Big[ \sum_{d=0}^{D/2-1} q_{[2d:2d+1]}^{\mathbb{C}} \cdot (k_{[2d:2d+1]}^{\mathbb{C}})^* \cdot e^{i(m-n)\theta_d} \Big],
\tag{2}
$$

where the operation $*$ represents the complex conjugation and $\cdot$ is the multiplication in the complex field. Expanding the right hand side of (2) and recovering the $m, n$ upper indices, we get

$$
\begin{aligned}
\text{Attn}_{\text{RoPE}}(q^{(m)}, k^{(n)}) =& \frac{1}{\sqrt{D}} \sum_{d=0}^{D/2-1} \Big( (q_{2d}^{(m)} k_{2d}^{(n)} + q_{2d+1}^{(m)} k_{2d+1}^{(n)}) \cdot \cos 2\pi(m-n)\theta_d \\
& + (q_{2d}^{(m)} k_{2d+1}^{(n)} - q_{2d+1}^{(m)} k_{2d}^{(n)}) \cdot \sin 2\pi(m-n)\theta_d \Big) \\
=:& \frac{1}{\sqrt{D}} \sum_{d=0}^{D/2-1} \Big( A_d^{(m,n)} \cos 2\pi(m-n)\theta_d + B_d^{(m,n)} \sin 2\pi(m-n)\theta_d \Big).
\end{aligned}
\tag{3}
$$

Here we adopt the following handy notations:

$$
\begin{aligned}
A_d^{(m,n)} &=: q_{2d}^{(m)} k_{2d}^{(n)} + q_{2d+1}^{(m)} k_{2d+1}^{(n)}, \\
B_d^{(m,n)} &=: q_{2d}^{(m)} k_{2d+1}^{(n)} - q_{2d+1}^{(m)} k_{2d}^{(n)}.
\end{aligned}
\tag{4}
$$

### 2.2 DISTANCE BIAS

It has been widely observed that RoPE attention decays as relative distance increases (Xiong et al., 2023; Sun et al., 2022). We give theoretical explanations for this phenomenon. Let us begin with an asymptotic result.

---

[1]Some literature adopt the notation "$b$" for base frequency $1/\theta_0$ and refer to $1/\theta_d$ as the wavelength of RoPE's $d$-th dimension. To simplify notations, we adopt "$\theta_0$" and avoid "$b$". Increasing RoPE base frequency is simply equivalent to decreasing $\theta_0$.

**Assumption 2.1.** There exists $\mu_0$, $\nu_0$, and $\sigma_0$, such that for all $m, n, d$ we have

$$\mathbb{E}_{q,k} A_d^{(m,n)} \equiv \mu_0, \quad \mathbb{E}_{q,k} B_d^{(m,n)} \equiv \nu_0, \quad \mathbb{E}_{q_d,k_d} |q_d^{(m)} k_d^{(n)}|^2 \equiv \sigma_0^2. \tag{5}$$

**Theorem 2.1** (Long-Context Instability). *Given Assumption 2.1 and let $\mu_0, \nu_0$ be as in* (5). *If $\{\theta_d\}_{d=1}^{D/2-1}$ are $\mathbb{Q}$-linear independent, then for any $\gamma \in [-\sqrt{\mu_0^2 + \nu_0^2}, \sqrt{\mu_0^2 + \nu_0^2}]$, there exists $\{(m_k, n_k)\}_k$ such that*

$$\limsup_{k \to \infty} \left| \frac{2}{\sqrt{D}} \mathbb{E}_{q^{(m_k)}, k^{(n_k)}} Attn_{RoPE}(q^{(m_k)}, k^{(n_k)}) - \gamma \right| = \mathcal{O}(\frac{1}{D}). \tag{6}$$

**Remark** The $\mathbb{Q}$-linear independence condition is satisfied when $\theta_0$ is transcendental, or when $\theta_0$ is algebraic but $\theta_0^{2/D}$ has degree higher than $D/2 - 1$.

Theorem 2.1 reveals an inherent instability in RoPE — its attention carries a token-agnostic, distance-dependent bias which varies so drastically over long range that it sub-converges to any value within a fixed interval up to an error of size $1/D$. Next, we establish a quantitative characterization of the distance bias of RoPE.

**Theorem 2.2.** *Given Assumption 2.1, there exists $\theta(\mu_0, \nu_0)$, such that if $\theta_0 < \theta(\mu_0, \nu_0)$, then for any $1 < |m - n| < \theta_0^{-\frac{1}{4}}/8$ and $|m' - n'| > \theta_0^{-1}$, we have*

$$\frac{sgn(\mu_0)}{\sqrt{D}} \left( \mathbb{E}_{q^{(m)}, k^{(n)}} Attn_{RoPE}(q^{(m)}, k^{(n)}) - \mathbb{E}_{q^{(m')}, k^{(n')}} Attn_{RoPE}(q^{(m')}, k^{(n')}) \right) > |\mu_0|/8 \tag{7}$$

*for sufficiently large $D$. Here $\mu_0, \nu_0$ are defined in* (5).

Theorem 2.2 asserts that when $\mu_0 > 0$ and context length is roughly $\mathcal{O}(\theta_0^{-1})$, RoPE attention favors nearby tokens more than distant ones, with a definite amount of attention score gap on average. This attention gap makes long-context modeling more challenging, because the model must expend extra capacity and computation to overcome the intrinsic bias against distant tokens and recover relevant long-range information. Fortunately, the next Theorem says that one can reduce such gap between any given position-pairs by further decreasing RoPE's $\theta_0$. This aligns with the approaches of various RoPE extension methods based on increasing base frequency $1/\theta_0$ (Xiong et al., 2023), or equivalently, performing positional interpolation (Chen et al., 2023).

**Theorem 2.3.** *Given Assumption 2.1, for any $\epsilon > 0$ and $m, n, m', n' > 0$, the following holds*

$$\frac{1}{\sqrt{D}} \left| \mathbb{E}_{q^{(m)}, k^{(n)}} Attn_{RoPE}(q^{(m)}, k^{(n)}) - \mathbb{E}_{q^{(m')}, k^{(n')}} Attn_{RoPE}(q^{(m')}, k^{(n')}) \right| < \epsilon \tag{8}$$

*for sufficiently small $\theta_0$ and large $D$.*

We introduce notations necessary for proofs in Appendix A, and present the proofs of Theorem 2.1, 2.2, 2.3 in Appendix B, C, and D respectively.

## 3 A NEW POSITIONAL ENCODING METHOD: TOKEN-AWARE PHASE ATTENTION (TAPA)

RoPE's distance bias is harmful for long-context language modeling, as it hurts model's ability in feeling long-range dependencies and leveraging distant information. We propose a new positional encoding paradigm to eliminate the undesired distance bias and preserve interactions with arbitrarily far-away tokens.

**Definition 3.1** (TAPA). Let $q, k$ be representation vectors of query and key located at position $m$ and $n$, $\phi(q, k)$ be any smooth function on the Cartesian space of $(q, k)$. Furthermore, let $\alpha$ be a positive real number, $D$ be the transformer head dimension, and $\mathcal{M} \in \mathbb{R}^{D \times D}$ be any square matrix. Then TAPA associated to $(\phi, \mathcal{M}, \alpha)$ is given by

$$Attn_{\phi, \mathcal{M}, \alpha}\left(q, k\right) = q^\top \mathcal{M} k \cdot \cos\left(2\pi |m - n|^\alpha \phi(q, k)\right). \tag{9}$$

Note TAPA reduces to the standard inner product attention when $\mathcal{M} = I_D$ and $\phi \equiv 0$. Among the many possible choices of $\phi$, we focus on the quadratic form[2]

$$\phi(q, k) = q^\top \mathcal{N} k, \quad \mathcal{N} \in \mathbb{R}^{D \times D}, \tag{10}$$

not only for its simplicity and better expressivity over linear functions, but more importantly because it offers the simplest *stationary phase* (Stein & Murphy, 1993) for suitable choices of $\mathcal{N}$ — one that possesses a single non-degenerate critical point. In Subsection 4.5, we compare quadratic phases with linear phases.

To further simplify TAPA, we segment query and key each into two parts

$$q = (q_A, q_P), \quad k = (k_A, k_P) \tag{11}$$

such that $q_A, k_A \in \mathbb{R}^{\theta D}$ and $q_P, k_P \in \mathbb{R}^{(1-\theta)D}$ for some hyperparameter $\theta \in (0, 1)$, and define

$$\mathrm{Attn}_{\theta,\alpha}\Big(q, k\Big) = \frac{q_A^\top k_A}{\sqrt{\theta D}} \cdot \cos\Big(2\pi |m - n|^\alpha \frac{q_P^\top k_P}{\sqrt{(1-\theta)D}}\Big). \tag{12}$$

Note this is a special case of (9) by setting

$$\mathcal{M} = \frac{1}{\sqrt{\theta D}} \cdot \begin{pmatrix} \mathbf{I}_{\theta D} & \mathbf{0} \\ \mathbf{0} & \mathbf{0} \end{pmatrix}, \quad \mathcal{N} = \frac{1}{\sqrt{(1-\theta)D}} \cdot \begin{pmatrix} \mathbf{0} & \mathbf{0} \\ \mathbf{0} & \mathbf{I}_{(1-\theta)D} \end{pmatrix} \tag{13}$$

Here the query and key subscripts "**A**" and "**P**" stand for "**A**mplitude" and "**P**hase" respectively. The hyperparameter $\theta$ controls how parameters are allocated between these two components. One notable benefit of the form (12) is that no new parameters or flops are introduced in addition to those of a vanilla transformer.

In contrast to RoPE's distance bias, whose limit points (as $|m - n| \to \infty$) are $\mathcal{O}(1/D)$-dense (see Theorem 2.1), TAPA's distance bias admits a unique limit of 0:

**Theorem 3.2** (Decaying Bias). *Let $\rho$ be the joint density function of $q_A$, $k_A$, $q_P$, $k_P$, and assume $\rho$ to be Schwartz class. Then there exists $C(\rho, D) > 0$, such that for $m \neq n$ we have*

$$\Big|\mathbb{E}_{q,k} \mathrm{Attn}_{\theta,\alpha}(q^{(m)}, k^{(n)})\Big| < C(\rho, D) \cdot |m - n|^{-\alpha(1-\theta)D}. \tag{14}$$

The rapid decay (14) is a result of cancellation in an *oscillatory integral* induced by the token-aware quadratic phase. Importantly, *pointwise* attention values need not converge to zero at all. In fact, the next Theorem establishes a lower bound on the variance of TAPA, showing that TAPA stays non-degenerate as distance grows, and is hence able to maintain interactions with arbitrarily distant tokens.

**Theorem 3.3** (Long-Context Non-Degeneracy). *Under Assumption 2.1 and let $\sigma_0^2$ be as in (5), we have*

$$\liminf_{|m-n| \to \infty} Var_{q,k}(Attn_{\theta,\alpha}(q^{(m)}, k^{(n)})) \geq \frac{\sigma_0^2}{2}. \tag{15}$$

The proofs of Theorem 3.2 and 3.3 are deferred to Appendix G and H. Lastly, we design experiments to visualize the distance bias of RoPE and TAPA, and obtain empirical evidence that TAPA is far less affected by distance bias than RoPE. See Appendix I for details.

## 4 EXPERIMENTS

We pretrain Transformers with LLaMA3 7B architecture on 8k context length, fine-tune on 32k, and evaluate up to 64k. Our experiments show that TAPA is able to adapt to 32k by only fine-tuning on less than $0.25\%$ of pretraining tokens, and extrapolate significantly better to the unseen length of 64k compared to all other baselines.

---

[2]For convenience, we refer to (10) as a quadratic form, as it is equivalent to the quadratic form defined on Cartesian space of $(q, k)$ given by $\frac{1}{2} \begin{pmatrix} q^\top & k^\top \end{pmatrix} \begin{pmatrix} 0 & \mathcal{N} \\ \mathcal{N}^\top & 0 \end{pmatrix} \begin{pmatrix} q \\ k \end{pmatrix}$.

### 4.1 PRETRAINING

For fair comparison, we pretrain Transformers with LLaMA3 7B (Dubey et al., 2024) architecture from scratch with TAPA (3.1) and RoPE (Su et al., 2021) respectively.

TAPA's formula (12) requires two $QK^\top$ dot products, which falls outside the standard SDPA form implemented by off-the-shelf FlashAttention kernels (Dao et al., 2022). We therefore implement TAPA in PyTorch without flash-style fusion. For a fair comparison, we use the same (non-flash) implementation for RoPE. While a flash-style implementation is feasible—e.g., via PyTorch FlexAttention with an extra $QK^\top$ matmul or a custom Triton kernel, these engineering focus is beyond the scope of this work and we leave it for future study.

For TAPA's implementations, we make no changes to transformer architecture other than removing RoPE and replacing transformer's inner product attention with Equation (12) where $\alpha = 0.1, \theta = 0.5$. For RoPE pretraining, we chose base frequency $1/\theta_0 = 5 \times 10^5$ as suggested by (Xiong et al., 2023).

The pretraining uses Pile (Gao et al., 2020), and each training document is chunked into 8k length segments. The pretraining uses $512 \times$ H100 GPUs with a global batch size of 256 for a total of 200k steps, which results in a total of 420B tokens. We use AdamW (Loshchilov & Hutter, 2017) with $\beta_1 = 0.9, \beta_2 = 0.95, \epsilon = 10^{-8}$ and no weight decay. The optimizer linearly warms up from 0 to the maximum learning rate in 5k steps and then decays according to a cosine schedule to $0.1\times$ maximum learning rate. We use $10^{-4}$ as the maximum learning rate for RoPE, while for TAPA we find that a smaller learning rate $2 \times 10^{-5}$ to be suitable.

### 4.2 LONG-CONTEXT FINE-TUNING

To extend to long context, we further fine-tune pretrained models with different Positional Encoding methods on the training split of PG19 (Rae et al., 2019), where each document is chunked into segments of length 32k. We fine-tune with each Positional Encoding method using a global batch size of 128 for 500 steps in total. The optimizer configuration is mostly the same as in pretraining, except that we use $2 \times 10^{-5}$ as the maximum learning rate across all methods, and warm up for only 50 steps.

For RoPE we fine-tuned with two base frequencies $b = 1/\theta_0$. The first reuses the pretraining setting $b = 5 \times 10^5$, and the second adopts a larger $b = 2 \times 10^8$ to detect any additional benefit from further increasing $b$.

In TAPA fine-tuning, we keep all hyper-parameters, architectures, and attention computations the same from pre-training. This **aligns with the key motivation of TAPA's design** — to enable scaling to longer contexts through direct fine-tuning and, unlike the RoPE family, does not require any position-scaling or hyperparameter tuning post-pretraining.

For PI (Chen et al., 2023), we set the max $L' = 65536$ (i.e. 64k), which is the maximal length we will evaluate our models on. For the same reason we set the scale factor to 8 in YaRN (Peng et al., 2023c). When fine-tuning with the increased base frequency approach (Xiong et al., 2023) we experimented with several options ranging from $10^{-6}$ to $2 \times 10^{-9}$, and report the best result achieved when base equals to $4 \times 10^{-9}$.

### 4.3 LONG-CONTEXT EVALUATION

We evaluate all fine-tuned models on the test split of PG19 (Rae et al., 2019) which consist of mostly long sequence samples. To measure models' performance at different context lengths, we consider segmentation of each document with context window size varying from 1k to 64k in the dyadic fashion. For each context window, we closely follow the sliding window method from (Press et al., 2021) with stride = 256 to calculate the test loss.

| Context window size | 1024 | 2048 | 4096 | 8192 | 16384 | 32768 | 49152 | 65536 |
|---|---|---|---|---|---|---|---|---|
| RoPE (b=5e5) | 12.97 | 12.53 | 12.22 | 11.97 | 11.79 | 12.96 | 938.23 | 2280.16 |
| RoPE (b=2e8) | 13.00 | 12.54 | 12.23 | 11.98 | 11.80 | 12.96 | 942.94 | 2284.72 |
| PI | 12.99 | 12.54 | 12.23 | 11.98 | 11.80 | 12.97 | 939.17 | 2282.44 |
| YaRN | 13.05 | 12.60 | 12.29 | 12.03 | 11.85 | 12.16 | 322.14 | 1962.55 |
| TAPA | 13.04 | 12.62 | 12.30 | 12.07 | 11.83 | 11.74 | 11.67 | 11.75 |

Table 1: Test perplexities on PG19 test set of LLaMA3 7B transformers first pretrained on 8k context length and further fine-tuned on 32k, and evaluated on 1k∼64k.

| Context window size | 1024 | 2048 | 4096 | 8192 | 16384 | 32768 |
|---|---|---|---|---|---|---|
| RoPE (b=5e5) | 13.08 | 12.63 | 12.32 | 12.21 | 5878.17 | 16366.63 |
| YaRN | 13.09 | 12.64 | 12.33 | 12.22 | 5869.35 | 16342.28 |
| PI | 13.07 | 12.62 | 12.30 | 12.19 | 5872.29 | 16350.28 |
| TAPA | 13.12 | 12.66 | 12.34 | 12.22 | 17.96 | 122.71 |

Table 2: Test perplexity via directly evaluating LLaMA3 7B transformers pretrained on 8k context length to 1k∼32k, without finetuning on 32k.

## 4.4 EVALUATION RESULTS

We report the test perplexity for multiple Positional Encoding methods on context window sizes ranging from 1k to 64k on the checkpoints obtained from Subsection 4.2.

As shown in Table 1, on short to mid context lengths (1k–16k) all position encodings exhibit a similar, monotonically decreasing perplexity curve (from ∼13.0 at 1k to ∼11.8 at 16k), with differences within a few hundredths. At 32k, a noticeable difference appears: TAPA attains the lowest perplexity (11.74), followed by YaRN (12.16), while RoPE/PI plateau around 12.96–12.97. Beyond this point, the trends diverge sharply. At 49k–64k, RoPE/PI and YaRN collapse. The perplexities blows up to ∼938–2285 (RoPE/PI) and ∼322–1963 (YaRN)—whereas TAPA remains stable (11.67 at 49k, 11.75 at 64k). In other words, while YaRN is more resilient than RoPE/PI it still collapses at very long lengths. TAPA is the only method that preserves low perplexity across the entire 1k–64k range, demonstrating substantially stronger long-context robustness and utilization than the alternatives.

In addition, we consider zero-shot long-context perplexity without 32k fine-tuning. We directly evaluate LLaMA3 7B models pretrained at 8k on 1k–32k and present the results in Table 2. It shows that all position encodings behave similarly on 1k–8k (perplexity decreases from ∼13.1 at 1k to ∼12.2 at 8k with sub-tenth differences). However, extrapolation beyond 8k fails for RoPE/PI/YaRN: at 16k their perplexities jump to around $5.87 \times 10^3$, and at 32k to roughly $1.63 \times 10^4$. In contrast, TAPA degrades gracefully, reaching 17.96 at 16k and 122.71 at 32k, which is about $327\times$ and $133\times$ lower than the next-best baseline, respectively. These results indicate that without any long-context fine-tuning, TAPA retains substantially better long-range generalization relative to all other baselines.

## 4.5 ABLATIONS: TAPA'S PHASE CHOICE

We compare two phase functions for TAPA: (i) *quadratic* (stationary) phase in (10) and (ii) *linear* (non-stationary) phase:

$$\phi(q,k) = \frac{1}{\sqrt{(1-\theta)D}} \cdot (q^\top, k^\top) \cdot \mathbb{1}_{(1-\theta)D}. \tag{16}$$

| Context window size | 1024 | 2048 | 4096 | 8192 | 16384 | 32768 | 49152 | 65536 |
|---|---|---|---|---|---|---|---|---|
| Linear Phase | 14.63 | 14.15 | 13.83 | 13.54 | 13.29 | 13.13 | 13.59 | 14.82 |
| Quadratic phase | 13.04 | 12.62 | 12.30 | 12.07 | 11.83 | 11.74 | 11.67 | 11.75 |

Table 3: Test perplexities on PG19 test set of LLaMA3 7B transformers first pretrained on 8k context length and further fine-tuned on 32k, and evaluated on 1k∼64k.

According to Table 3, TAPA with quadratic phase consistently outperforms the linear variant across all lengths. In the short–mid range (1k–16k), quadratic improves from $13.04 \to 11.83$ versus $14.63 \to 13.29$ for linear—an absolute gap of $1.3-1.6$ ($\approx 11\%$ relative at 1k and 16k). At longer lengths the divergence grows and stability differs markedly: at 32k, linear reaches 13.13 while quadratic is 11.74 ($\Delta = 1.39$, $\approx 11\%$); beyond 32k the linear curve becomes non-monotonic and degrades, whereas quadratic remains flat and low. These results align with the intuitions from the theoretical perspective of oscillatory integrals, where non-stationary phases (e.g., linear) induce large, rapidly varying oscillations in representation space, while stationary phases are less sensitive to small representation changes.

However, it is worth noting that although TAPA with linear phase is suboptimal compared to the quadratic phase, it still dominates the baselines in Table 1 at long ranges, achieving a significantly lower orders of magnitude: e.g., at 49k/64k it attains $13.59/14.82$ perplexity versus $322-943$ and $1963-2285$ for YaRN and RoPE/PI, respectively.

Overall, the stationary (quadratic) phase yields both better accuracy and greater long-context stability, while even the linear-phase TAPA retains strong long-context robustness compared to RoPE family.

# 5 RELATED WORK

## 5.1 POSITIONAL ENCODING IN TRANSFORMERS

Positional encodings in transformers were originally introduced as fixed sinusoidal embeddings (Vaswani et al., 2023), and later extended to learned embeddings (Radford & Narasimhan, 2018; Radford et al., 2019; Devlin et al., 2019). These absolute position methods inject positional signals independent of token content and is limited by fixed context length. Relative position encodings (Shaw et al., 2018; Dai et al., 2019; Raffel et al., 2019; He et al., 2020; Huang et al., 2020; Ke et al., 2020; Press et al., 2021) address this by letting model attend to relative distances typically as additive biases in the attention scores. CAPE (Likhomanenko et al., 2021) on the other hand augments absolute sinusoidal positional embeddings with randomized continuous shifts and scaling during training.

## 5.2 ROTARY POSITIONAL EMBEDDING (ROPE) AND EXTENSIONS

Rotary Position Embedding (RoPE) (Su et al., 2021) offers a novel way to encode relative positions by applying complex-valued rotations directly to the representations of queries and keys. However, the original RoPE does poorly in directly extrapolating to long context, leading to various studies and extension methods to mitigate this issue.

XPos (Sun et al., 2022) adds an exponential scaling on top of RoPE to enforce a more explicit preference based on token's relative positions and empirically smooths attention oscillations compared to RoPE.

However, it is observed (Xiong et al., 2023) that XPos with hyperparameters unchanged after pretraining still suffer from the same long-context bottleneck as RoPE and they have to apply a base frequency change to XPos to fix the issue. In fact, the authors of (Xiong et al., 2023) hypothesize that the bottleneck of extrapolation ability of both RoPE and XPos lies in the heavy decay of attention score as relative distance grows; to mitigate such decay they propose to increase RoPE's base frequency, and observed better adaptation on longer context. In (Liu et al., 2023) the authors further study the scaling law of different choices of base frequencies.

An insightful and popular concurrent approach at the time is Position-Interpolation (PI) (Chen et al., 2023), which linearly rescale long context positions to interpolate pretraining position range. Their hypothesis is that directly extrapolating to any unseen position could lead to unstable attention behavior, as in theory the trigonometric terms in RoPE's formula is able to approximate any value (when RoPE's dimension is sufficiently large). It is worth noting that position interpolation is mathematically equivalent to increasing RoPE's base frequency non-uniformly across RoPE dimensions, although they stem from different hypotheses and motivations.

YaRN (Peng et al., 2023c) hypothesizes that applying the same position rescaling or enlarged base frequency to all RoPE dimensions may cause token representations to become closer to each other and make it harder for model to differentiate and understand local relationships; they propose a new interpolation scheme that determines interpolation strength across RoPE dimensions in a non-linear fashion.

Similarly, LongRoPE (Ding et al., 2024) manage to extend to 2 million context length via a non-uniform rescaling that differentiates higher frequency dimensions from lower frequency ones. They apply minimal scalings to early tokens and propose an efficient search algorithm to find optimal scaling configurations.

All these methods rely on heuristic rescaling of input positions or manual hyperparameter adjustments after pre-training in order to extend context length. The rescaling functions are often ad hoc and hyperparameter choices are subtle and case-by-case, typically requiring extensive ablation to identify suitable configurations. In addition, some prior work has also included theoretical analyses of RoPE, such as bounding RoPE attention scores from above (Su et al., 2021), or an interpolation error bound of Positional Interpolation (Chen et al., 2023), but they do not explain the limitation of RoPE or why interpolation helps.

## 5.3 NON-RoPE APPROACHES TO POSITIONAL EXTRAPOLATION

While not being the focus of this work, it is worth mentioning several non-RoPE extrapolation approaches. NoPE (Haviv et al., 2022; Chi et al., 2023; Kazemnejad et al., 2023) proposes to remove any explicit use of positional encoding from transformers as they hypothesize that the asymmetry of causal mask in language modeling already implicitly encodes positional information. Nevertheless, the limitations of NoPE in scaling to long contexts have been demonstrated both empirically and theoretically (Wang et al., 2024; Ma et al., 2024). LM-Infinite (Han et al., 2023) introduces a Lambda-shaped masking mechanism to enable extrapolation that is applicable to any relative positional encoding; hence it is orthogonal to our approach as we preserve the causal masking mechanism and focus on improving the positional encoding itself. Another line of research explores alternative designs of the attention mechanism, incorporating custom positional encoding schemes. Examples include FNet (Lee-Thorp et al., 2021), LongNet (Fu et al., 2023), RWKV (Peng et al., 2023a), and Hyena (Poli et al., 2023). Since these methods deviate from the standard transformer architecture, whereas our work assumes the traditional attention, we do not discuss them in depth.

## 6 CONCLUSION

We introduced TAPA, a new positional–encoding paradigm which, unlike RoPE, requires *no* hyperparameter changes or input re-scalings after pretraining, and adapts to longer context range with only light fine-tuning.

On the theory side, we proved that RoPE carries an intrinsic distance bias in attention, which limits RoPE's long context ability, and we provided a mathematical justification for various RoPE extensions including Position-Interpolation (PI). On the other hand, we established vanishing bias and non-degeneracy guarantees for TAPA as context grows.

Empirically, on LLaMA3 7B models pretrained at 8k and evaluated up to 64k, TAPA matches strong baselines at short ranges, and is the only method that continues to improve up to ∼49k, and remains stable at 64k, while RoPE-family methods collapse. Zero-shot tests (no 32k fine-tuning) further confirm TAPA's robustness at long range.

## LIMITATION

Our study targets language models and primarily evaluates with perplexity. Extending TAPA to larger model scales, multimodalities, and evaluation metrics beyond perplexity (e.g., long-form QA, code, and reasoning) is a natural next step. Experiment-wise, among the various positional encoding strategies, we focus our baselines and theoretical studies primarily on RoPE and its variants, because of RoPE's unique role and widespread usage in modern open-source Large Language Models (LLMs). On the modeling side, exploring phase families beyond the quadratic form, and scaling to 128k–1M token regimes are promising directions. From an engineering perspective, TAPA introduces additional phase/amplitude computations (effectively two extra quadratic operations per head), which can be slower than vanilla attention without further optimizations such as using FlashAttention-like kernels. Overall, TAPA offers a principled and practical path toward reliable long-context modeling.

## ETHICS STATEMENT

We use established research datasets for training and evaluation and do not collect human-subject data. We do not target or analyze personal information, and all experiments are offline; no end-user deployment is involved. While improved long-context modeling can benefit benign uses (e.g., summarization, retrieval, and analysis), it may also increase the ability to generate longer harmful content; downstream applications should pair our method with standard safety measures (filtering, rate limits, and alignment). We report key training details to support reproducibility. No sensitive attributes were intentionally modeled, but inherited dataset biases may persist.

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

## A  PREPARATIONS FOR PROVING THEOREM 2.1, 2.2, AND 2.3

Consider the following decomposition of (3):

$$
\frac{1}{\sqrt{D}}\text{Attn}_{\text{RoPE}}(q^{(m)}, k^{(n)}) = \left( \frac{\mu_0}{D} \cdot \sum_{d=0}^{D/2-1} \cos 2\pi(m-n)\theta_d + \frac{\nu_0}{D} \cdot \sum_{d=0}^{D/2-1} \sin 2\pi(m-n)\theta_d \right)
$$
$$
+ \frac{1}{D}\left( \sum_{d=0}^{D/2}(A_d - \mu_0)\cos 2\pi(m-n)\theta_d + \sum_{d=0}^{D/2}(B_d - \nu_0)\sin 2\pi(m-n)\theta_d \right) \tag{17}
$$
$$
=: \Gamma_\lambda + Z_\lambda
$$

where $\lambda = m - n$. Note that $\mathbb{E}_{q,k} Z_\lambda = 0$ by definition (5), and LHS of (7) is precisely $\Gamma_\lambda - \Gamma_\Lambda$. For convenience, we denote it by:

$$
\Delta_{\lambda,\Lambda} =: \Gamma_\lambda - \Gamma_\Lambda. \tag{18}
$$

**Remark**  In addition to Assumption 2.1, if we further assume that $\{A_i\}_i$ and $\{B_j\}_j$ are sub-gaussian satisfying

$$
\mathbb{P}(|A_i - \mu_0| > \eta) < C_1 e^{-C_2 \eta^2}, \quad \mathbb{P}(|B_j - \nu_0| > \eta) < C_1 e^{-C_2 \eta^2} \tag{19}
$$

for some $C_1, C_2 > 0$, $\mu_0, \nu_0 \in \mathbb{R}$, all $\eta > 0$ and $i, j = 0, \cdots, D/2 - 1$, then one can show[3] that the non-deterministic term $Z_\lambda - Z_\Lambda$ admits the following control

$$
\mathbb{P}(|Z_\lambda - Z_\Lambda| > \eta) < C_3 e^{-C_4 D \eta^2} \tag{20}
$$

for some $C_3, C_4 > 0$ and all $\eta > 0$. In other words, the magnitude of $Z_\lambda - Z_\Lambda$ is negligible (with high probability) compared to $\Delta_{\lambda,\Lambda}$ when $D$ is large.

---

[3]For example, using Laplace transform.

## B    PROOF OF THEOREM 2.1

*Proof of Theorem 2.1.* A direct computation of the expectation of $\text{Attn}_{\text{RoPE}}(q^{(m)}, k^{(n)})$ gives:

$$\frac{2}{\sqrt{D}}\mathbb{E}_{q^{(m)}, k^{(n)}}\text{Attn}_{\text{RoPE}}(q^{(m)}, k^{(n)}) =: \frac{2}{D}\sum_{d=0}^{D/2-1}\left(\mu_0\cos 2\pi(m-n)\theta_d + \nu_0\sin 2\pi(m-n)\theta_d\right) \tag{21}$$

$$= \frac{2\sqrt{\mu_0^2 + \nu_0^2}}{D}\sum_{d=0}^{D/2-1}\sin\left(2\pi(m-n)\theta_d + \phi\right)$$

where $\phi = \arctan(\mu_0/\nu_0)$. Define $\beta = \arcsin(\gamma/\sqrt{\mu_0^2 + \nu_0^2})/2\pi - \phi \mod 1$. Since $\{\theta_d\}_{d=1}^{D/2-1}$ are $\mathbb{Q}$-linearly independent, it follows from Weyl's criterion that $\left((\lambda\theta_1), \cdots, (\lambda\theta_{D/2-1})\right)$, $\lambda = 1, 2, \cdots$ is uniformly distributed in $[0,1]^{D/2-1}$ (e.g. Theorem 6.3 and Example 6.1 in Kuipers & Niederreiter (1974)). Here $(r)$ represents the fractional part of a real number $r$. Consequently, for any $\epsilon > 0$ we can find $\lambda_\epsilon \in \mathbb{Z}$ sufficiently large such that

$$\left|\left((\lambda_\epsilon\theta_1), \cdots, (\lambda_\epsilon\theta_{D/2-1})\right) - (\beta, \cdots, \beta)\right| \tag{22}$$

is so small that

$$\left|\frac{2}{D}\sum_{d=0}^{D/2-1}\sin\left(\lambda_\epsilon\theta^{2d/D} + \phi\right) - \frac{\gamma}{\sqrt{\mu_0^2 + \nu_0^2}}\right| < \epsilon + \mathcal{O}(\frac{1}{D}). \tag{23}$$

Repeating this argument on a sequence $\epsilon_k \to 0$ leads to the conclusion of Theorem 2.1. $\qquad\square$

## C    PROOF OF THEOREM 2.2

The proof of Theorem 2.2 employs estimates of the following two sums:

$$\mathcal{C}_D(\lambda) =: \frac{1}{D}\sum_{d=0}^{D/2-1}\cos 2\pi\lambda\theta_0^{2d/D},$$

$$\mathcal{S}_D(\lambda) =: \frac{1}{D}\sum_{d=0}^{D/2-1}\sin 2\pi\lambda\theta_0^{2d/D}. \tag{24}$$

**Lemma C.1.** *Given $\theta_0 < 1/10$, $D > 4|\log\theta_0|$, and $\lambda > 1$, then the following inequalities hold:*

$$|\mathcal{C}_D(\lambda)| \le \frac{2}{\theta_0|\log\theta_0|\lambda\pi} + \epsilon(D; \lambda, \theta_0, \alpha),$$

$$|\mathcal{S}_D(\lambda)| \le \frac{2}{|\log\theta_0|} + \epsilon(D; \lambda, \theta_0, \alpha), \tag{25}$$

*for all $\alpha > 0$, where*

$$\epsilon(D; \lambda, \theta_0, \alpha) =: \alpha + \frac{4\pi\lambda\theta_0^\alpha}{D}. \tag{26}$$

**Lemma C.2.** *Assume $\theta_0 < 1/10$, $D > 4|\log\theta_0|$, and $\lambda > 1$. If $\lambda\theta_0^{\epsilon_0} < 1/4$ for some $\epsilon_0 > 0$, then we have*

$$\mathcal{C}_D(\lambda) > \frac{1}{2}\cdot(1-\epsilon_0)\cdot\cos 2\pi\lambda\theta_0^{\epsilon_0} - \frac{1}{|\log\theta_0|} - \epsilon(D; \lambda, \theta_0, \alpha), \tag{27}$$

*for all $\alpha > 0$ where $\epsilon(D; \lambda, \theta_0, \alpha)$ is defined in (26).*

The proofs of Lemma C.1 and C.2 will be given in Appendix E and F.

*Proof of Theorem 2.2.* Choosing $\epsilon_0 = 1/4$ and using $\lambda\theta_0^{1/4} < 1/8$, it follows from Lemma C.2 that

$$
\begin{aligned}
\mathcal{C}_D(\lambda) &> \frac{1}{2} \cdot \frac{3}{4} \cdot \frac{\sqrt{2}}{2} - \frac{2}{|\log\theta_0|} - \epsilon(D; \lambda, \theta_0, \alpha) \\
&> \frac{1}{4} - \frac{2}{|\log\theta_0|} - \epsilon(D; \lambda, \theta_0, \alpha).
\end{aligned}
\tag{28}
$$

On the other hand, using $\Lambda > \theta_0^{-1}$ and Lemma C.1, we have

$$
\mathcal{C}_D(\Lambda) < \frac{2}{|\log\theta_0|} + \epsilon(D; \Lambda, \theta_0, \alpha).
\tag{29}
$$

In addition, the following is a direct consequence of Lemma C.1:

$$
\begin{aligned}
\mathcal{S}_D(\lambda) &< \frac{2}{|\log\theta_0|} + \epsilon(D; \lambda, \theta_0, \alpha), \\
\mathcal{S}_D(\Lambda) &< \frac{2}{|\log\theta_0|} + \epsilon(D; \Lambda, \theta_0, \alpha).
\end{aligned}
\tag{30}
$$

Combining (28), (29), and (30), using the definition (5), we bound $\text{sgn}(\mu_0)\Delta_{\lambda,\Lambda}$ (where $\Delta_{\lambda,\Lambda}$ is defined in (18)) from below as follows:

$$
\begin{aligned}
\text{sgn}(\mu_0)\Delta_{\lambda,\Lambda} &\geq \mathcal{C}_D(\lambda) \cdot |\mu_0| - |\mathcal{C}_D(\Lambda) \cdot \mu_0| - |\mathcal{S}_D(\lambda) \cdot \nu_0| - |\mathcal{S}_D(\Lambda) \cdot \nu_0| \\
&> \frac{|\mu_0|}{4} - \frac{4(|\mu_0| + |\nu_0|)}{|\log\theta_0|} - 2(|\mu_0| + |\nu_0|) \cdot \Big(\epsilon(D; \lambda, \theta_0, \alpha) + \epsilon(D; \Lambda, \theta_0, \alpha)\Big).
\end{aligned}
\tag{31}
$$

By first choosing $\theta_0$ such that $\theta_0 < \exp(-64(|\mu_0| + |\nu_0|)/|\mu_0|)$, we have

$$
\frac{4(|\mu_0| + |\nu_0|)}{|\log\theta_0|} < \frac{|\mu_0|}{16},
\tag{32}
$$

and then increasing $D$ such that

$$
2(|\mu_0| + |\nu_0|) \cdot \Big(\epsilon(D; \lambda, \theta_0, \alpha) + \epsilon(D; \Lambda, \theta_0, \alpha)\Big) < \frac{|\mu_0|}{16}
\tag{33}
$$

for some properly chosen $\alpha$ (e.g. $\alpha = 0$), we arrive at

$$
\text{sgn}(\mu_0)\Delta_{\lambda,\Lambda} > \frac{|\mu_0|}{4} - \frac{|\mu_0|}{16} - \frac{|\mu_0|}{16} = \frac{|\mu_0|}{8}.
\tag{34}
$$

Thus Theorem 2.2 is concluded. □

## D   PROOF OF THEOREM 2.3

*Proof.* By definition of $\mathcal{C}_D(\lambda)$ from (24), we have the trivial bound

$$
\mathcal{C}_D(\lambda) < \frac{1}{2}
\tag{35}
$$

holds for all $\lambda$. Now choose $\theta_0$ to be sufficiently small such that

$$
\max(\lambda\theta_0^{\epsilon_0}, \Lambda\theta_0^{\epsilon_0}) < \frac{1}{4},
\tag{36}
$$

for some $\epsilon_0$ to be determined later. Then using (35) and (36), we can apply Lemma C.2 to see that

$$
\begin{aligned}
\Delta_{\lambda,\Lambda} &< \frac{|\mu_0|}{2} - \frac{|\mu_0|}{2} \cdot (1 - \epsilon_0) \cdot \cos 2\pi\Lambda\theta_0^{\epsilon_0} - \epsilon(D; \Lambda, \theta_0, \alpha)|\mu_0|, \\
-\Delta_{\lambda,\Lambda} &< \frac{|\mu_0|}{2} - \frac{|\mu_0|}{2} \cdot (1 - \epsilon_0) \cdot \cos 2\pi\lambda\theta_0^{\epsilon_0} - \epsilon(D; \lambda, \theta_0, \alpha)|\mu_0|.
\end{aligned}
\tag{37}
$$

By first choosing $\epsilon_0 < \frac{\epsilon}{4}$, then further decreasing $\theta_0$ such that

$$
\begin{aligned}
|\mu_0| \cos 2\pi \Lambda \theta_0^{\epsilon_0} &> |\mu_0| - \frac{\epsilon}{4}, \\
|\mu_0| \cos 2\pi \lambda \theta_0^{\epsilon_0} &> |\mu_0| - \frac{\epsilon}{4},
\end{aligned}
\tag{38}
$$

and lastly increasing $D$ such that

$$
\begin{aligned}
\epsilon(D; \lambda, \theta_0, \alpha)|\mu_0| &< \epsilon/4, \\
\epsilon(D; \Lambda, \theta_0, \alpha)|\mu_0| &< \epsilon/4,
\end{aligned}
\tag{39}
$$

we obtain

$$
|\Delta_{\lambda, \Lambda}| < \epsilon,
\tag{40}
$$

which proves Theorem 2.3. $\qquad\square$

**Remark** One can directly verify Theorem 2.3 using the following elementary facts

$$
\begin{aligned}
\lim_{\theta_0 \to 0} \mathcal{C}_D &= (\cos 2\pi\lambda + D/2 - 1)/D, \\
\lim_{\theta_0 \to 0} \mathcal{S}_D &= \sin 2\pi\lambda/D.
\end{aligned}
\tag{41}
$$

Namely, choose a sufficiently large $D$ such that $4/D < \epsilon/2$, and then choose $\theta_0$ sufficiently small. But such argument lacks a quantitative understanding of the limiting behavior and the relation among the variables in question. We adopt an alternative proof above using Lemma C.2 to explicitly quantify the smallness of $\theta_0$ in terms of $\lambda$ and $\epsilon$.

# E    PROOF OF LEMMA C.1

*Proof of Lemma C.1.* Without ambiguity and for simplicity, drop $\lambda$ from the expressions of $\mathcal{C}_D(\lambda), \mathcal{S}_D(\lambda)$ throughout the proof. First let us focus on $\mathcal{C}_D$. By treating $\mathcal{C}_D$ as a Riemann sum we can rewrite it as follows:

$$
\begin{aligned}
\mathcal{C}_D &= \left( \frac{1}{D} \sum_{d=0}^{D/2} \cos 2\pi\lambda \theta_0^{2d/D} - \frac{1}{2} \int_0^1 \cos 2\pi\lambda \theta_0^x dx \right) + \frac{1}{2} \int_0^1 \cos 2\pi\lambda \theta_0^x dx \\
&= \frac{1}{2} \sum_{d=0}^{D/2} \int_{2d/D}^{2(d+1)/D} \left( \cos 2\pi\lambda \theta_0^{2d/D} - \cos 2\pi\lambda \theta_0^x \right) dx + \frac{1}{2} \int_0^1 \cos 2\pi\lambda \theta_0^x dx \\
&=: \frac{1}{2}\Delta + \frac{1}{2}\mathcal{I}.
\end{aligned}
\tag{42}
$$

We first consider $\Delta =: \sum_{d=0}^{D/2} \Delta_d$. For arbitrary $\alpha \in (0,1)$ we may split the sum into two parts:

$$
\Delta = \sum_{d \leq \alpha D/2} \Delta_d + \sum_{d > \alpha D/2} \Delta_d =: \Delta' + \Delta''.
\tag{43}
$$

For each term in $\Delta'$, we use the fact that cosine functions are bounded by 1 and control it as follows:

$$
|\Delta_d| \leq \frac{2}{D} \cdot 2 = \frac{4}{D}.
\tag{44}
$$

For each term in $\Delta''$, we instead use the Lipschitz bound of the integrand:

$$
|\Delta_d| \leq \mathrm{Lip}_d \cdot \frac{4}{D^2},
\tag{45}
$$

where

$$
\mathrm{Lip}_d =: \sup_{[\frac{2d}{D}, \frac{2(d+1)}{D}]} |\frac{d}{dx} \cos 2\pi\lambda \theta_0^x| = 2\pi\lambda |\log \theta_0 \cdot \theta_0^x \sin 2\pi\lambda \theta_0^x| \leq 2\pi\lambda |\log \theta_0| \theta_0^{2d/D}.
\tag{46}
$$

Now plugging both (44) and (45) into (43), we obtain:

$$|\Delta'| \leq \frac{D\alpha}{2} \cdot \frac{4}{D} = 2\alpha,$$

$$
\begin{aligned}
|\Delta''| &\leq \frac{4}{D^2} \cdot 2\pi\lambda |\log\theta_0| \cdot \sum_{d=D\alpha/2+1}^{D/2} \theta_0^{2d/D} \\
&\leq \frac{4}{D^2} \cdot 2\pi\lambda |\log\theta_0| \cdot \theta_0^{\frac{2}{D}(\frac{D\alpha}{2}+1)} \cdot \frac{1}{1-\theta_0^{2/D}} \\
&\leq \frac{8\pi\lambda |\log\theta_0|}{D^2} \cdot \theta_0^{\alpha} \cdot \left(\frac{1}{1-\theta_0^{2/D}} - 1\right).
\end{aligned}
\tag{47}
$$

Using 2nd order Taylor's expansion with remainder of Lagrange form, we obtain that

$$1 - \theta_0^{2/D} \geq |\log\theta_0| \cdot \frac{2}{D} - |\log\theta_0|^2 \cdot \frac{4}{D^2}. \tag{48}$$

Here we have used the following fact:

$$\sup_{x\in[0,\frac{2}{D}]} \left|\frac{d^2}{dx^2}\theta_0^x\right| \leq |\log\theta_0|^2. \tag{49}$$

Now inserting (49) into the estimate of $\Delta''$ in (47), we get

$$|\Delta''| \leq \frac{4\pi\lambda}{D} \cdot \theta_0^{\alpha} \cdot \frac{1}{1-2|\log\theta_0|/D} \leq \frac{4\pi\lambda}{D} \cdot \theta_0^{\alpha} \cdot \frac{1}{1-1/2} = \frac{8\pi\lambda\theta_0^{\alpha}}{D}. \tag{50}$$

We have now arrived at the bound for the first term in (43):

$$\left|\frac{\Delta}{2}\right| \leq \alpha + \frac{4\pi\lambda\theta_0^{\alpha}}{D} =: \epsilon(D; \lambda, \theta_0, \alpha), \tag{51}$$

where $\epsilon(D; \lambda, \theta_0, \alpha)$ is defined in (26) and $\alpha \in (0,1)$ is arbitrary. Next, we estimate the integral term in (42). By performing a change of variable $y = \theta_0^x$ we see that

$$\frac{1}{2} \cdot \mathcal{I} = \frac{1}{2|\log\theta_0|} \int_{\theta_0}^{1} \frac{\cos 2\pi\lambda y}{y} dy = \frac{1}{2|\log\theta_0|} \int_{\lambda\theta_0}^{\lambda} \frac{\cos 2\pi y}{y} dy. \tag{52}$$

Note the right hand side of (52) is an oscillatory integral, so we may employ the cancellation effect to control it. Define

$$
\begin{aligned}
n_0 &= \min\{n \in \mathbb{Z} : \lambda\theta_0 \leq \frac{1}{4}(4n+1)\}, \\
N_0 &= \max\{n \in \mathbb{Z} : \lambda \geq \frac{1}{4}(4n+5)\}.
\end{aligned}
\tag{53}
$$

Decompose the integration interval of (52) as follows (for brevity we omit the integrand):

$$
\begin{aligned}
&\int_{\lambda\theta_0}^{\frac{1}{4}(4n_0+1)} + \sum_{n=n_0}^{N_0} \left(\int_{\frac{1}{4}(4n+1)}^{\frac{1}{4}(4n+3)} + \int_{\frac{1}{4}(4n+3)}^{\frac{1}{4}(4n+5)}\right) + \int_{\frac{1}{4}(4N_0+5)}^{\lambda} \\
&= I_* + \sum_{n=n_0}^{N_0} (I_n^- + I_n^+) + I^*.
\end{aligned}
\tag{54}
$$

According to (53), integrals $I_*$ and $I^*$ contain at most a full period of $\cos 2\pi y$, and therefore can be trivially bounded:

$$|I^*| + |I_*| \leq \frac{1}{\lambda\theta_0} + \frac{1}{\lambda - 1} \leq \frac{2}{\lambda\theta_0}. \tag{55}$$

Here we used the assumption that $\theta_0 < 1/10$ and $\lambda > 1$ in the second inequality above. For the sum term in the middle, we have

$$|I_n^- + I_n^+| = |I_n^-| - |I_n^+| \leq \frac{4}{4n+1} \cdot \frac{1}{\pi} - \frac{4}{4n+5} \cdot \frac{1}{\pi} \leq \frac{1}{\pi} \cdot \frac{1}{n^2}. \tag{56}$$

Here we used the fact that $\cos 2\pi y$ is constantly non-positive on the integral range of $I_n^-$, and therefore

$$|I_n^-| = -\int_{\frac{4n+1}{4}}^{\frac{4n+3}{4}} \frac{\cos 2\pi y}{y} dy \le -\frac{4}{4n+1} \int_{\frac{4n+1}{4}}^{\frac{4n+3}{4}} \cos 2\pi y dy = \frac{4}{4n+1} \cdot \frac{1}{\pi}. \tag{57}$$

Similarly, $\cos 2\pi y$ is constantly non-negative on the integral range of $I_n^+$, and thus

$$-|I_n^+| = -\int_{\frac{4n+3}{4}}^{\frac{4n+5}{4}} \frac{\cos 2\pi y}{y} dy \le -\frac{4}{4n+5} \int_{\frac{4n+3}{4}}^{\frac{4n+5}{4}} \cos 2\pi y dy = \frac{4}{4n+5} \cdot \frac{1}{\pi}. \tag{58}$$

so the sum admits the following bound:

$$\sum_{n=n_0}^{N_0} (I_n^- + I_n^+) \le \frac{1}{\pi} \sum_{n=n_0}^{N_0} \frac{1}{n^2} \le \frac{1}{\pi} \cdot \frac{1}{n_0 - 1} \le \frac{1}{\pi} \cdot \frac{2}{\lambda \theta_0}, \tag{59}$$

where we used the first identity in (53). Inserting (55) and (59) into (52), we have

$$\frac{1}{2} \cdot \mathcal{I} \le \frac{2}{\theta_0 |\log \theta_0| \lambda \pi}. \tag{60}$$

Combining (51) and (60), we concluded the estimate of $\mathcal{C}_D$ in (25). Next we estimate $\mathcal{S}_D$. We point out that most the proofs of bounding $\mathcal{S}_D$ follows the same line as that of $\mathcal{C}_D$, so to avoid repetitive argument, therefore we state without proving all results that are achievable through same techniques as its $\mathcal{C}_D$ counterpart, and only focus on addressing the difference.

First, we conduct a similar decomposition of $\mathcal{S}_D$ as (42), into $\frac{\Delta}{2} + \frac{\mathcal{I}}{2}$. The estimate of $\Delta$ follows from exactly the same lines as that of $\mathcal{C}_D$, hence we omit the details:

$$\frac{\Delta}{2} \le \epsilon(D; \lambda, \theta_0, \alpha). \tag{61}$$

To estimate $\frac{\mathcal{I}}{2}$, we again use the change of variable $y = \theta_0^x$ and similar to (52) we get

$$\frac{\mathcal{I}}{2} = \frac{1}{2|\log \theta_0|} \int_{\lambda \theta_0}^{\lambda} \frac{\sin 2\pi y}{y} dy. \tag{62}$$

To bound this oscillatory integral we adopt the following decomposition of integral region:

$$\int_{\lambda \theta_0}^{1/2} + \sum_{n=0}^{N_0} \left( \int_{\frac{1}{2}(2n+1)}^{\frac{1}{2}(2n+2)} + \int_{\frac{1}{2}(2n+2)}^{\frac{1}{2}(2n+3)} \right) + \int_{\frac{1}{2}(2N_0+3)}^{\lambda}$$

$$= I_* + \sum_{n=0}^{N_0} (I_n^- + I_n^+) + I^*, \tag{63}$$

where

$$N_0 = \max\{n \in \mathbb{Z} : \lambda \ge \frac{1}{2}(2n+3)\}. \tag{64}$$

Note again that the integrand is non-positive in $I_n^-$, and non-negative in $I_n^+$. Similar to (55) we have

$$|I_n^- + I_n^+| \le \frac{1}{\pi} \cdot |-\frac{2}{2n+1} + \frac{2}{2n+3}| = \frac{1}{\pi} \cdot \frac{4}{(2n+1)(2n+3)} < \frac{1}{\pi} \cdot \frac{1}{n^2}, \tag{65}$$

and therefore

$$\sum_{n=0}^{N_0} (I_n^- + I_n^+) < \frac{1}{\pi} \sum_{n=0}^{N_0} \frac{1}{n^2} < \frac{\pi}{6} < 1. \tag{66}$$

Next, we use the fact that $\sin 2\pi y/y$ is bounded by $1$ on the interval $[0, 1/2]$ to trivially bound $I_*$:

$$|I_*| \leq \frac{1}{2}. \tag{67}$$

The integral in the last term contains at most a full period, and thus admits the following bound:

$$|I^*| < \frac{1}{\lambda - 1} \leq 1. \tag{68}$$

Combining (66), (67), and (68), we have

$$\frac{\mathcal{I}}{2} < \frac{2}{|\log \theta_0|}. \tag{69}$$

Lastly, combining (61) and (69), we proved the estimates of $\mathcal{S}_D$ in (25). Thus we concluded Lemma C.1. □

## F    PROOF OF LEMMA C.2

*Proof of Lemma C.2.*  The proof reuses the decomposition (42) and the bound (51), but further needs a lower bound for $\mathcal{I}/2$. First we decompose the right hand side integral of (52) as follows:

$$\frac{1}{2|\log \theta_0|} \left( \int_{\lambda\theta_0}^{1/4} + \int_{1/4}^{\lambda} \right) \frac{\cos 2\pi y}{y} dy. \tag{70}$$

Following the same argument to bound the middle term in (66), we have

$$\left| \frac{1}{2|\log \theta_0|} \int_{1/4}^{\lambda} \frac{\cos 2\pi y}{y} dy \right| < \frac{1}{2|\log \theta_0|} \cdot 2 = \frac{1}{|\log \theta_0|}. \tag{71}$$

Next, notice that the integrand stays positive on $[\lambda\theta_0, 1/4]$, we hence have

$$\frac{1}{2|\log \theta_0|} \int_{\lambda\theta_0}^{1/4} \frac{\cos 2\pi y}{y} dy > \frac{1}{2|\log \theta_0|} \int_{\lambda\theta_0}^{\lambda\theta_0^{\epsilon_0}} \frac{\cos 2\pi y}{y} dy > \frac{\cos 2\pi\lambda\theta_0^{\epsilon_0}}{2|\log \theta_0|} \cdot \int_{\lambda\theta_0}^{\lambda\theta_0^{\epsilon_0}} \frac{dy}{y}$$
$$= \frac{\cos 2\pi\lambda\theta_0^{\epsilon_0}}{2|\log \theta_0|} \cdot |\log \theta_0|(1 - \epsilon_0) = \frac{1}{2} \cdot (1 - \epsilon_0) \cdot \cos 2\pi\lambda\theta_0^{\epsilon_0}. \tag{72}$$

Finally, combining (51), (71), and (72), we obtain

$$|\mathcal{C}_D| \geq \frac{1}{2}\mathcal{I} - \frac{1}{2}|\Delta| \geq \frac{1}{2|\log \theta_0|} \int_{\lambda\theta_0}^{1/4} \frac{\cos 2\pi y}{y} dy - \left| \frac{1}{2\log \theta_0} \int_{1/4}^{\lambda} \frac{\cos 2\pi y}{y} dy \right| - \frac{1}{2}|\Delta|$$
$$\geq (1 - \epsilon_0) \cdot \cos 2\pi\lambda\theta_0^{\epsilon_0} - \frac{1}{|\log \theta_0|} - \epsilon(D; \lambda, \theta_0, \alpha), \tag{73}$$

which is exactly (27), and hence proved Lemma C.2. □

## G    PROOF OF THEOREM 3.2

*Proof of Theorem 3.2.*  For convenience, we introduce the following notations:

$$x_A := (q_A, k_A), \quad x_P := (q_P, k_P). \tag{74}$$

First let us expand the expression of the expectation of TAPA:

$$\int_{x_A} \frac{x_A^\top \cdot J_{\theta D} \cdot x_A}{\sqrt{\theta D}} \left( \int_{x_P} \cos \left( \frac{2\pi|m - n|^\alpha}{\sqrt{(1 - \theta)D}} \cdot x_P^\top \cdot J_{(1-\theta)D} \cdot x_P \right) \cdot \rho(x_A, x_P) dx_P \right) dx_A \tag{75}$$

where $J_d = \begin{pmatrix} \mathbf{0} & I_d \\ I_d & \mathbf{0} \end{pmatrix}$. Let us further simplify the expression by writing $\lambda =: \frac{2|m-n|^\alpha}{\sqrt{(1-\theta)D}}$. The inner integral of (75) can now written as

$$\int_{x_P} \cos\left(\pi\lambda \cdot x_P^\top \cdot J \cdot x_P\right) \cdot \rho(x_A, x_P) dx_P = \text{Re}\left(\int_{x_P} e^{-i\pi\lambda \cdot x_P^\top \cdot J \cdot x_P} \cdot \rho(x_A, x_P) dx_P\right). \tag{76}$$

where we omitted the subscript $(1-\theta)D$ of $J$ when there is no ambiguity present. Applying Fourier Transform to imaginary Gaussian (e.g. Proposition 6.2 in (Wolff, 2003)), we manage to bound the integral on the right hand side of (76) as follows:

$$\left|\int_{x_P} e^{-i\pi\lambda \cdot x_P^\top \cdot J \cdot x_P} \cdot \rho(x_A, x_P) dx_P\right| \leq C\lambda^{-(1-\theta)D}\left(\sup_{x_P}|\rho(x_A, x_P)| + \lambda^{-2}\sum_{|\alpha_P|=2}\sup_{x_P}|D^{\alpha_P}\rho(x_A, x_P)|\right), \tag{77}$$

where the summation is taken over all second order derivatives with respect to the $x_P$ variable, and $C$ is a universal constant. Since $\rho$ is in Schwartz class, the following seminorms of $\rho$ admit fast decay in $x_A$:

$$\sup_{x_P}|\rho(x_A, x_P)|, \quad \sup_{x_P}|D^{\alpha_P}\rho(x_A, x_P)|. \tag{78}$$

Therefore the following function is integrable in $x_A$:

$$\Phi_\lambda(x_A) =: \frac{x_A^\top \cdot J_{\theta D} \cdot x_A}{\sqrt{\theta D}} \cdot \left(\sup_{x_P}|\rho(x_A, x_P)| + \lambda^{-2}\sum_{|\alpha_P|=2}\sup_{x_P}|D^{\alpha_P}\rho(x_A, x_P)|\right). \tag{79}$$

Note when $|m-n| \neq 0$, by definition we have $\lambda \geq C(D) > 0$. Therefore $\{\Phi_\lambda\}_\lambda$ is uniformly bounded in $L^1$:

$$\left|\int_{x_A}\Phi_\lambda(x_A)dx_A\right| \leq C(\rho, D). \tag{80}$$

Combining (75), (77), and (80), we have shown that

$$\left|\mathbb{E}_{q,k}\text{Attn}_{\theta,\alpha}(q^{(m)}, k^{(n)})\right| \leq C(\rho, D)\lambda^{-(1-\theta)D} = C'(\rho, D)|m-n|^{-\alpha(1-\theta)D}. \tag{81}$$

Thus proves Theorem 3.2. $\qquad\square$

# H  PROOF OF THEOREM 3.3

*Proof.* By exploiting the elementary identity $\cos^2 x = (1 + \cos 2x)/2$, we may expand the second moment of Attention as follows:

$$\begin{aligned}
\mathbb{E}\left|\text{Attn}_{\theta,\alpha}\right|^2 &= \frac{1}{2}\int_{x_A}\frac{(x_A^\top \cdot J_{\theta D} \cdot x_A)^2}{\theta D}dx_A \\
&\quad + \frac{1}{2}\int_{x_A}\frac{x_A^\top \cdot J_{\theta D} \cdot x_A}{\sqrt{\theta D}}\left(\int_{x_P}\cos\left(\frac{4\pi|m-n|^\alpha}{\sqrt{(1-\theta)D}} \cdot x_P^\top \cdot J_{(1-\theta)D} \cdot x_P\right) \cdot \rho(x_A, x_P) dx_P\right)dx_A \\
&= I + II.
\end{aligned} \tag{82}$$

Notice that

$$I = \frac{1}{2\theta D}\mathbb{E}_{q_A,k_A}|q_A^\top k_A|^2 = \frac{\sigma_0^2}{2} \tag{83}$$

where we used Assumption 2.1 in the second identity and $\sigma_0^2$ is defined in 5. Next, we follow an exactly identical argument of estimating $\mathbb{E}\text{Attn}_{\theta,\alpha}$ (see Appendix G) to obtain

$$|II| \leq C''(D, \rho)|m-n|^{-\alpha(1-\theta)D}. \tag{84}$$

Lastly, combining (81), (83), (84), and taking $\liminf$ as $|m-n| \to \infty$, we proved (15), which concludes Theorem 3.3. $\qquad\square$

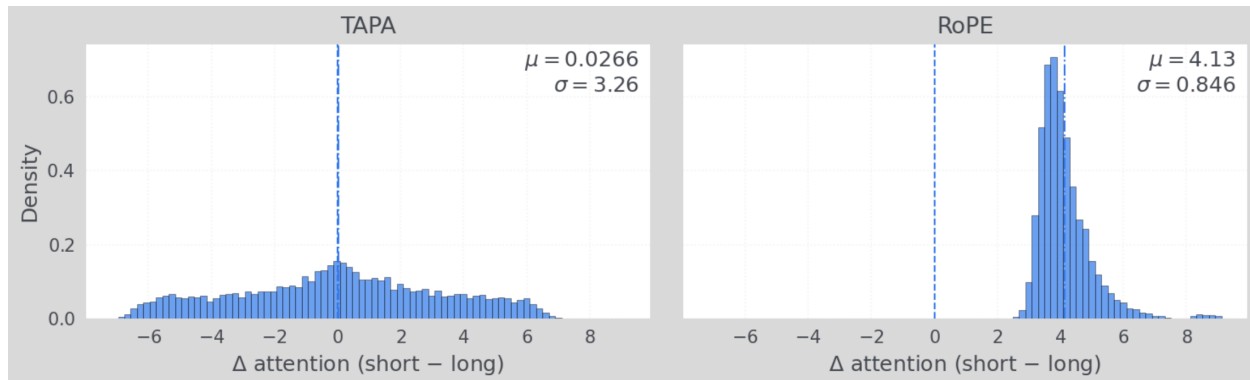

Figure 1: Empirical distributions of attention score differences, computed over 10000 randomly sampled pairs with positions drawn from $[0, 100]$ (short range) and $[10000, 10100]$ (long range). The skewed distribution of RoPE reflects its strong distance bias, whereas the near-symmetric distribution of TAPA indicates that no significant bias is present.

## I VISUALIZE DISTANCE BIAS

To visuallize the distance bias of RoPE and TAPA, we compare the distributions of their attention scores difference between short-range and long-range token pairs. More precisely, given two disjoint intervals $I_{\text{short}} = [0, 100]$ and $I_{\text{long}} = [10000, 10100]$, we randomly sample $\lambda \in I_{\text{short}}$ and $\Lambda \in I_{\text{long}}$, and compute the difference

$$\Delta = \text{Attn}_\lambda - \text{Attn}_\Lambda.$$

We randomly sample 10000 pairs of attention scores satisfying the above positional constraint from the evaluation results on PG19 testset. The resulting histograms (Figure 1) show that TAPA produces a highly symmetric distribution centered near zero ($\mu \approx 0.03, \sigma \approx 3.3$), which indicate low distance bias between near and far token-pairs, while RoPE yields a distribution shifted significantly toward positive values ($\mu \approx 4.1, \sigma \approx 0.85$), revealing a systematic bias that favors short-range interactions.

