# OpenReview forum: "Positional Encoding via Token-Aware Phase Attention"
_ICLR.cc/2026/Conference — Submitted to ICLR 2026_

### Official Review · Reviewer_8R3a · 2025-10-23

**Soundness:** 3
**Presentation:** 3
**Contribution:** 3
**Rating:** 6
**Confidence:** 3

**Summary:**

This paper introduces Token-Aware Phase Attention (TAPA), a novel positional encoding method to deal with the limitations of Rotary Positional Embedding (RoPE) in long-context language modeling. The authors first provide a theoretical proof that RoPE has an intrinsic distance-dependent bias, which limits its ability to long-range modeling. TAPA overcomes this by incorporating a learnable phase function into the attention mechanism, which eliminates this bias. The paper demonstrates theoretically that TAPA's distance bias vanishes as context grows while maintaining non-degenerate attention. Empirically, the authors pretrain a LLaMA3 7B model and show that TAPA significantly outperforms RoPE and its variants (like PI and YaRN) on long-context perplexity benchmarks, especially when extrapolating to unseen context lengths. When pretrained on 8k context without finetuning, TAPA remains effective up to 32k context length, while other methods see a collapse in performance.

**Strengths:**

* The paper presents a mathematical proof of the inherent distance-dependent bias in RoPE. It also provides theoretical guarantees for TAPA, proving its decaying distance bias and non-degeneracy for long contexts.

* TAPA is a well-motivated idea to the prevalent RoPE-based methods. Unlike many RoPE extensions that rely on heuristic adjustments such as position rescaling, TAPA is presented as a more fundamental solution that can be extended to longer contexts with minimal fine-tuning and without manual hyperparameter changes.

* The head-to-head comparison with RoPE, PI, and YaRN on a LLaMA3 7B architecture demonstrates TAPA's superior performance in long-context scenarios. In particular, under a 8k context pretrain setting, TAPA remains stable up to 32k context length while the other methods collapse.The zero-shot evaluation also highlights TAPA's generalization ability.

**Weaknesses:**

* TAPA's formulation requires two QK dot products, which is computationally more expensive than the standard attention mechanism. While they mention the feasibility of a custom implementation, the practical overhead in terms of training and inference speed compared to highly optimized RoPE implementations is a potential limitation.

* The experiments are conducted on a 7B LLaMA3 architecture. Demonstrating the effectiveness of TAPA on a wider range of model sizes and architectures would strengthen the generalizability of the claims.

* The evaluation is focused on perplexity. Although this is a standard metric, it may not fully capture all aspects of long-context understanding, such as reasoning over long documents or complex instruction following.

**Questions:**

* Could you elaborate on the practical implications of the increased computational cost of TAPA? Have you performed any analysis on the slowdown compared to a flash-attention-based RoPE implementation during training and inference?

* The paper focuses on a quadratic phase function for TAPA. While the ablation shows it outperforms a linear phase, have you explored other families of functions for the learnable phase?

* Do you have insights into how pre-training with TAPA from scratch might differ from pre-training with RoPE in terms of training dynamics or final model capabilities?

---

> ### Author Response · Authors · 2025-11-25
> **Author Response to Reviewer 8R3a, part I**
>
> We sincerely thank the reviewer for the thoughtful and supportive assessment. We greatly appreciate your detailed reading of the submission and your constructive comments, which have helped us clarify several aspects of TAPA’s formulation and evaluation. Below we address each point in turn.
>
> **Weakness 1**: Thank you for raising this concern. First we clarify that TAPA does not double attention cost: the two QK$^\top$ dot products operate on disjoint sub-dimensions whose total size is still $D$, so the overall attention FLOPs remain $O(L_q L_k D)$, matching standard attention in leading order.
> To assess the practical overhead, we implemented a FlashAttention-style streaming kernel in Python (blockwise online softmax, no $L\times L$ materialization) for both standard attention and TAPA. Although this prototype is slower than a fused CUDA/Triton kernel, it reliably measures relative slowdown.
> Across sequence lengths from 8k to 32k, we observed only a $1.1$–$1.3\times$ slowdown—far below the $2\times$ cost of a naive unfused implementation.
> A fused Triton/CUDA kernel would stream $q_A,q_P,k_A,k_P$ once, accumulate both dot products within the same tile, and apply the phase modulation before softmax, sharing memory loads and reductions. Thus, the real overhead should remain close to FlashAttention, with only a modest constant factor.
> We will release the Python reference implementation used for these comparisons in the final version. Overall, TAPA preserves the same asymptotic complexity as standard attention and remains compatible with high-performance fused kernels.
>
> **Weakness 2**: We appreciate the suggestion to evaluate more architectures. In this work, we used the LLaMA3 7B architecture purely as a template: every model is randomly initialized and pretrained from scratch, with no use of open-source checkpoints. This allows us to isolate the effect of the positional encoding itself without confounding from inherited representations.
> Pretraining a 7B model from scratch requires $32\times 8$ H100 GPUs for nearly 12 days, and our available compute budget did not permit scaling to larger models. We will clarify this computational limitation in the revision.
>
> **Weakness 3**: We fully agree that downstream long-context skills such as long-range reasoning and complex instruction following are crucial.
> However, our study focuses strictly on pretraining from scratch only on PG19 dataset. Without a more thorough pretraining, post-training or instruction-tuning phase, the models lack the generative and instruction-following capabilities required to run such benchmarks. Our evaluation therefore follows standard practice in architectural/positional-encoding research at pretraining time: measuring perplexity and long-context stability directly, so that effects can be attributed purely to the positional encoding rather than downstream supervision.
>
> **Question 1**: Thank you for this question. A naive, unfused implementation indeed incurs overhead because it materializes two $L\times L$ attention matrices. However, this is an implementation artifact—not an intrinsic limitation of the method.
> To measure intrinsic cost, we implemented a FlashAttention-style streaming kernel in Python. Across 8k–32k context lengths, the slowdown remained only $1.1$–$1.3\times$. This matches the behavior of a fused kernel, where:
> $q_A,q_P,k_A,k_P$ are streamed once,
> both dot products are accumulated in the same tile, and
> the phase modulation is applied before softmax with no extra memory movement.
> We will release the reference implementation for transparency, and will clarify in the paper that optimized kernels can keep TAPA very close to FlashAttention in practice.
>
> **Question 2**: We appreciate this insightful question. From a harmonic analysis viewpoint, linear and quadratic phases are the simplest representatives of the two major oscillatory families: non-stationary (linear) and stationary (quadratic). They require the least parameters and computational cost in their own categories, making them the most natural starting point for introducing token-dependent phase functions in attention.
> In theory, higher-order stationary phases share a similar qualitative stability and oscillation behavior as the quadratic form, but with a natural expressivity–stability trade-off. Exploring richer phase families is an interesting future direction, and we will emphasize this in the revision.

---

> ### Author Response · Authors · 2025-11-25
> **Author Response to Reviewer 8R3a, part II**
>
> (Continuing from part I)
>
> **Question 3**: Although a full mechanistic analysis is outside the scope of this paper, our theory yields several insights.
> By the RoPE gap theorem (Theorem 2.2), RoPE inherently favors short-range interactions when context length approaches $O(\theta^{-1})$. During pretraining, this induces a strong locality bias, and we observe increasingly anisotropic long-range attention patterns, along with gradient instability when extending to very long contexts.
> TAPA, by contrast, eliminates intrinsic distance bias: its positional effect arises only through token semantics. As a result, long-range interactions remain stable, and optimization proceeds more smoothly on long sequences.
> These differences are reflected empirically in our extension experiments, where RoPE-based models degrade beyond 8k–16k, whereas TAPA maintains stable performance to 32k.
>
> **Lastly**, thank you again for the encouraging review and helpful questions. We truly appreciate your positive assessment and will incorporate these clarifications to further strengthen the paper.

---

> > ### Author Response · Authors · 2025-11-27
> > **Response to reviewer 8R3a, follow up on weakness 3**
> >
> > We agree with the reviewer that perplexity alone cannot fully capture all aspects of long-context understanding. While our models are pretrained from scratch without any instruction tuning and therefore cannot directly evaluate complex generative tasks, we can still meaningfully evaluate long-range retrieval ability.
> >
> > Motivated by the reviewer’s suggestion to look beyond perplexity, and in line with a related comment from Reviewer 6Cy2, we extended our evaluation to include the single-needle-in-a-haystack (NIAH) benchmark. Our original models were trained only on PG19, which consists primarily of long-form narrative text and contains very limited question–answer structure, factual key–value patterns, or numerical copy-style supervision. As a result, these models tended to default to continuation rather than extractive retrieval, making NIAH evaluation unreliable in the initial submission.
> >
> > To make such evaluation meaningful, we performed continual pretraining of all pretrained checkpoints (32k max length) with an additional 5B tokens drawn from the following open-source mixture:
> > - 30% BookCorpusOpen
> > - 30% Wikipedia
> > - 30% C4
> > - 10% StackExchange
> >
> > Each component contributes complementary capabilities for long-range retrieval: BookCorpusOpen provides long-form narrative structure; Wikipedia introduces dense key–value and factual patterns; C4 adds diverse web-style noise robustness; and StackExchange supplies number- and symbol-rich QA text. Together, we expect these sources to fill the gaps left by PG19-only pretraining.
> >
> > For NIAH evaluation, we followed the setup in the RULER paper (similar to Table 11). We constructed haystacks from concatenated Paul Graham essays, inserted a single magic-number needle at a random location, and appended the query at the end. For each PE method and each sequence length (1k–64k), we generated 100 independent examples and reported both per-length accuracy and the overall average. Results are shown below:
> >
> > | PE Method        | 1K  | 2K  | 4K  | 8K  | 16K  | 32K  | 64K | Avg. |
> > |------------------|-----|-----|-----|-----|------|------|-----|-------|
> > | RoPE (b=5e5)     | 99 | 100| 100 | 99 | 96 | 95   | 0   |  84.1    |
> > | RoPE (b=2e8)     | 98 | 97 | 99 | 99| 96 | 93 | 0   |  83.1    |
> > | PI               | 100 | 98 | 98 | 99 | 98 | 97 | 0   |   84.3   |
> > | YaRN             | 100 | 99 | 98 | 98 | 99 | 98 | 0   |   84.6   |
> > | **TAPA**         | 99 | 98 | 98 | 98 | 99  | **100**  | **96**  | **98.3**    |
> >
> > Notably, all existing baselines completely fail to extrapolate to the unseen length of 64k tokens, whereas TAPA maintains strong performance at this length. TAPA also achieves the highest accuracy at 32k, indicating improved long-range adaptability and retrieval capability while maintaining competitive short-range performance.
> >
> > We thank the reviewer again for highlighting the limitation of perplexity-only evaluation; this feedback motivated us to broaden our analysis and substantially strengthen both our empirical results and the clarity of the contribution. We hope the expanded evaluation helps convey a better picture of the model’s capabilities.

---

### Official Review · Reviewer_6Cy2 · 2025-10-28

**Soundness:** 2
**Presentation:** 3
**Contribution:** 2
**Rating:** 4
**Confidence:** 4

**Summary:**

This article re examines the shortcomings of remote attenuation in Rotary Positional Embedding (RoPE) and proposes a new position encoding called s Token-Aware Phase Attention (TAPA) based on this, which is beneficial for length extrapolation.

**Strengths:**

1.  Sufficient theoretical analysis. There is theoretical analysis on the weaknesses of ROPE's distance-dependent bias (Theorem 2.1, 2.2) and the relative advantages of TAPA (Theorem 3.2) in this regard.

2. Innovative solutions. A new solution (TAPA) for length extrapolation has been proposed and be verify in Section 4.3.

**Weaknesses:**

1.**Evaluation is limited**, without a long window benchmark, such as  a needle in a haystack [1] or ruler [2].

2.The assumption 2.1 **has not been verified**.  For example, the assumption of 2.1 may not be satisfied for any $d$.  As shown in [3][4], there is significant anisotropy between different dimensions.

3.Lack of comparison. Perhaps we can compare alibi (with different slopes) and sliding window attention (with different window size). There are other methods that do not rely on RoPEs and can be extrapolated, e.g. CAPE [5], antmax[6].


[1] Looking for a Needle in a Haystack: A Comprehensive Study of Hallucinations in Neural Machine Translation. https://aclanthology.org/2023.eacl-main.75/

[2] RULER: What’s the Real Context Size of Your Long-Context Language Models? https://openreview.net/forum?id=kIoBbc76Sy

[3]The Rotary Position Embedding May Cause Dimension Inefficiency in Attention Heads for Long-Distance Retrieval https://aclanthology.org/2025.findings-acl.697.pdf

[4]On the token distance modeling ability of higher RoPE attention dimension https://aclanthology.org/2024.findings-emnlp.338.pdf

[5]CAPE: Context-Adaptive Positional Encoding for Length Extrapolation https://arxiv.org/html/2405.14722v1

[6] Long-Context Generalization with Sparse Attention https://arxiv.org/pdf/2506.16640

**Questions:**

1.In my opinion, there is a **logical incompleteness** in this article, namely the unclear correlation between intrinsic distance dependent bias in attention scores and length extrapolation. In fact, when there is strong decay bias, the model can actually achieve better extrapolation on PPL, such as using swa [5], or even the a larger $\theta$ in RoPE [6].  **However**, this will present a superficial contextual ability, where the model's PPL can extrapolate, but in reality, it cannot perform tasks such as finding a needle in a haystack.

2.At the same time, even the a larger $\theta$ in RoPE [6] can lead to better length extrapolation. But in line 130 “Fortunately, the next Theorem says that one can reduce such gap between any given position-pairs by further decreasing RoPE’s $\theta_0$".  Can the contradiction presented by these two be explained using the theorem presented in this article?

3.There is no code. This article does not provide code for implementing TAPA, especially for efficiently utilizing the Triton/Telelang. I am currently unclear about its speed difference compared to RoPE.

4.Will partial ROPE[10], as a recently popular setting [11,12], affect the conclusions of this article?

5.In [13], the author of RoPE get context length $L$ scale with the $2 \theta_0^{-1}$  by using $Ci(x)$ under a sufficiently large $d$. This is consistent with line 126's statement that 'context length is $O(\theta_0^{-1})$ Is there a connection？

[7] Dissecting transformer length extrapolation via the lens of receptive field analysis. https://aclanthology.org/2023.acl-long.756.pdf

[8] Scaling laws of rope-based extrapolation. https://openreview.net/forum?id=JO7k0SJ5V6

[9] Base of RoPE Bounds Context Length. https://openreview.net/pdf?id=EiIelh2t7S

[10] https://github.com/lucidrains/x-transformers/issues/40

[11] DeepSeek-V2: A Strong, Economical, and Efficient Mixture-of-Experts Language Model https://arxiv.org/pdf/2405.04434

[12] Kimi K2: Open Agentic Intelligence https://arxiv.org/pdf/2507.20534

[13] https://kexue.fm/archives/10122

---

> ### Author Response · Authors · 2025-11-25
> **Author Response to Reviewer 6Cy2, part I**
>
> We thank the reviewer for the detailed and constructive feedback. We address each concern below and will clarify these points explicitly in the revised version.
>
> **Weakness 1**: We agree that long-window benchmarks such as “needle in a haystack’’ and “ruler’’ are important for evaluating post-trained generative models. However, our work focuses on pretraining from scratch only using PG19 training set and does not include a post-training or instruction-tuning stage. As a result, our models do not possess generative capabilities required to run these benchmarks. Instead, we follow the standard practice for architectural and positional-encoding studies at pretraining time: we evaluate perplexity and long-context stability directly during training. This isolates the effect of the positional encoding without confounding factors from downstream fine-tuning. We will clarify this scope in the revision.
>
> **Weakness 2**: We appreciate the reviewer’s concern regarding Assumption 2.1. We adopted Assumption 2.1 due to its highly simplified, coordinate-wise form solely for readability; our proof does not require such a strong restriction. In the revision, we will note that Assumption~2.1 can be replaced by a strictly weaker and more realistic boundedness condition:
>
> **Assumption**: Define
> \begin{eqnarray}
> \begin{split}
> \mu_{m,n} \coloneqq \frac{\mathbb{E}{q,k}\sum{d=0}^{D/2}A_d\cos(2\pi\lambda\theta_d)}{\sum_{d=0}^{D/2}\cos(2\pi\lambda\theta_d)},\quad
> \nu_{m,n} \coloneqq \frac{\mathbb{E}{q,k}\sum{d=0}^{D/2}B_d\sin(2\pi\lambda\theta_d)}{\sum_{d=0}^{D/2}\sin(2\pi\lambda\theta_d)}.
> \end{split}
> \end{eqnarray}
> There exist $C_0,c_0>0$ such that $|\mu_{m,n}|,|\nu_{m,n}|<C_0$ for all $m,n>0$, and $|\mu_{m,n}|>c_0$ for $|m-n|<\theta_0^{-1/4}/8$.
> This revised assumption does not require any per-dimension constancy or isotropy; it requires only boundedness of aggregated bilinear statistics, which is compatible with the anisotropy reported in prior work. We will (i) present this weaker assumption, (ii) provide empirical verification using our pretrained RoPE model on PG-19, and (iii) clarify that the original Assumption~2.1 was included solely to simplify arguments.
>
> **Weakness 3**: We agree that alternative positional encodings and architectural approaches exist, such as ALiBi, sliding-window attention, CAPE, and AntMax. However, as stated in the Introduction, this paper is specifically motivated by building a principled understanding of RoPE’s long-context behavior and proposing a remedy for its geometric limitations. RoPE remains the dominant positional encoding in modern open-source LLMs (e.g., LLaMA, Mistral, Qwen), and our contributions are aimed directly at addressing its known issues.
>
> That said, we did evaluate ALiBi in our setup. Its perplexity and long-context performance were **substantially worse** than all RoPE-based baselines at the 7B pretraining scale. We chose to not to include ALiBi in main tables as it would not affect the relative conclusions and would otherwise distract from the RoPE-focused comparison that this paper is designed to study. We will add a brief discussion of these results in the revision.
>
> In addition, methods such as sliding-window attention, CAPE, and AntMax address orthogonal concerns (e.g., complexity reduction, different extrapolation mechanisms) rather than positional geometry itself. They can in principle be combined with TAPA, but they are not direct substitutes for a RoPE-corrective positional encoding and therefore fall outside the primary comparison scope.
>
> **Question 1**: Thank you for raising this conceptual point. We recognize that the we didn't to explain our central principle sufficiently, and we will improve this in the revision. The central principle is that a positional encoding should not impose any intrinsic distance-dependent bias that persists after integrating over token semantics (i.e. taking expectations over all semantics). In other words, without context, the positional component should be position-independent or asymptotically stable. If the intrinsic bias depends on $|m-n|$, the attention mechanism is forced to prioritize or suppress interactions purely based on distance, which degrades modeling at both short and long ranges.
>
> Under this perspective, the link between intrinsic distance bias and length extrapolation is direct: non-vanishing intrinsic distance bias causes extrapolation degradation, while the absence of such bias enables stable long-range behavior. Our analysis shows that RoPE retains a non-vanishing intrinsic distance bias even after integrating over semantics (which subconverge to any number in a fixed interval according to Theorem 2.1), causing long-context deterioration. TAPA is explicitly designed so this intrinsic bias vanishes asymptotically: its token-aware phase ensures positional effects arise only through contextual content. We will clarify this connection in the revision.

---

> > ### Author Response · Authors · 2025-11-25
> > **Author Response to Reviewer 6Cy2, part II**
> >
> > (continuing from part I)
> >
> > **Question 2**: Thank you for noting this. This is indeed a terminology mismatch, which we should have made more explicit. In our paper, $\theta$ refers to the inverse of the RoPE frequency ($\theta = 1/\text{frequency}$) (see the footnote on page 2). Therefore, what the reviewer describes as “increasing the RoPE frequency’’ corresponds exactly to decreasing $\theta$ in our notation. Under this mapping, there is no contradiction. We will make this explicit in the revised text.
> >
> > **Question 3**: We appreciate the question about computational cost. A naive "unfused" implementation of the algorithm indeed leads to higher computation costs as it requires materializing two attention score matrices of size $L\times L$. However this is just an artifact of the under-optimized implementation, not an intrinsic drawback of the method. To evaluate this more carefully, we later implemented a FlashAttention-style streaming kernel in Python (blockwise online softmax, no $L\times L$ materialization) for both standard attention and TAPA. While this Python version is slower than a fused CUDA/Triton kernel, it is sufficient for measuring relative overhead. Across sequence lengths from 8k to 32k, we consistently observed only a $1.1$–$1.3\times$ slowdown—far below the $2\times$ overhead that a naive, unfused implementation would incur. This aligns with the expected behavior of a fused kernel, where:
> >
> > 1. $q_A,q_P,k_A,k_P$ are streamed once,
> > 2. both dot-product terms are accumulated in the same tile,
> > 3. the phase modulation is applied before softmax without extra memory movement.
> >
> > We will release the Python implementation for full transparency. In a proper fused Triton/CUDA kernel, memory reads and softmax reductions are shared, and runtime should remain very close to FlashAttention with only a modest constant factor.
> >
> > **Question 4**: Thank you for bringing this up. Techniques such as partial positional encodings function as architectural add-ons and can be applied on top of TAPA if desired. They are complementary rather than competitive, and therefore not directly comparable to TAPA as standalone positional encodings. Our focus is on improving the core positional encoding mechanism; integrating such add-ons with TAPA would be a promising direction for future work.
> >
> > **Question 5**: Yes there is a direct connection. This arises from approximating RoPE attention using the Riemann-sum expression as in Eq. (24). Lemma C.1 (Eq. 25) provides a bound for it where $\lambda$ represents token distance (i.e. effective context length). Morally speaking, when $\lambda \cdot \theta \sim \mathrm{const}$, the attention has a decent magnitude and positional bias does not dominate semantic contributions. However, as context length grows and $\lambda$ becomes large, the noisy $\varepsilon$ term in Eq.~25 becomes dominant, causing RoPE’s positional bias to grow disproportionately and lead to instability in long-context settings.
> >
> > Lastly, we sincerely appreciate the reviewer’s careful reading and hope that these clarifications resolve the concerns and help support a more positive evaluation.

---

> ### Comment · Reviewer_6Cy2 · 2025-11-25
> **Response to Author**
>
> Thanks for the rebuttal. My main concern is the following statement:
>
> >>Weakness 1: We agree that long-window benchmarks such as “needle in a haystack’’ and “ruler’’ are important for evaluating post-trained generative models. However, our work focuses on pretraining from scratch only using PG19 training set and does not include a post-training or instruction-tuning stage. As a result, our models do not possess generative capabilities required to run these benchmarks.
>
> Based on my personal experience and also the results in the "ruler" ( see Table 11 in https://arxiv.org/pdf/2404.06654), base model (  which is a 7B model trained with 420B tokens in your paper ) can also perform tasks such as finding a "needle in a haystack" or "ruler" without using SFT. So I strongly recommend the author to conduct a corresponding evaluation.

---

> > ### Author Response · Authors · 2025-11-27
> > **Response to reviewer 6Cy2's response (adding evaluation results for needle in haystack)**
> >
> > We thank the reviewer for pointing out that even base models can perform well on NIAH-style evaluations. Our original models were trained only on the PG19 corpus, which consists primarily of long-form narrative text and contains very limited question–answer structure, factual key–value or numerical copy-style patterns. Empirically, these pre-trained models tend to default to continuation behavior rather than extractive retrieval, which made NIAH evaluation unreliable in the initial submission.
> >
> > To properly fulfill the reviewer’s suggestion, we performed continual pretraining of all previous pre-trained checkpoints at the length of 32k with an additional 5B tokens using the following open-source mixture:
> >
> > - 30% BookCorpusOpen
> > - 30% Wikipedia
> > - 30% C4
> > - 10% StackExchange
> >
> > We chose above mixture because each source contributes a distinct capability required for long-range retrieval:
> > BookCorpusOpen provides long-form narrative structure, Wikipedia introduces dense key–value and factual patterns, C4 adds diverse web-style noise robustness, and StackExchange provides number- and symbol-rich question–answer data. Together, they fill the gaps left in PG19-only pretraining and we believe that makes NIAH evaluation meaningful.
> >
> > In our evaluation, we followed RULER's paper (similar setup as in Table 11) by constructing haystacks from concatenated Paul Graham essays and inserting a single magic-number needle at a random position of each haystack. For each PE method and each context length (dyadic-ally ranging from 1k to 64k), we generated 100 independent examples and reported the total number of successful retrievals, as well as the average accuracy across all lengths. The results are summarized below:
> >
> > | PE Method        | 1K  | 2K  | 4K  | 8K  | 16K  | 32K  | 64K | Avg. |
> > |------------------|-----|-----|-----|-----|------|------|-----|-------|
> > | RoPE (b=5e5)     | 99 | 100| 100 | 99 | 96 | 95   | 0   |  84.1    |
> > | RoPE (b=2e8)     | 98 | 97 | 99 | 99| 96 | 93 | 0   |  83.1    |
> > | PI               | 100 | 98 | 98 | 99 | 98 | 97 | 0   |   84.3   |
> > | YaRN             | 100 | 99 | 98 | 98 | 99 | 98 | 0   |   84.6   |
> > | **TAPA**         | 99 | 98 | 98 | 98 | 99  | **100**  | **96**  | **98.3**    |
> >
> > Notably, all existing baselines **completely failed** to extrapolate to the unseen length of 64k tokens, whereas TAPA **maintains strong performance** at this length. TAPA also achieves the highest accuracy at 32k, which proves its better long-range adaptability and retrieval capability while maintaining competitive short-range performance. We will include these results and clarifications in the revision.
> >
> > We again thank the reviewer for highlighting this direction, which enabled us to further improve the results. We hope the expanded evaluation and the strong performance of TAPA across long contexts will help the reviewer reassess the contribution more favorably.

---

> > > ### Comment · Reviewer_6Cy2 · 2025-11-27
> > > **Response to Author**
> > >
> > > My main concern has been resolved. I will increas the score. I hope the author can include the discussion of relevant literature and experimental results in the revised version.

---

> > > > ### Author Response · Authors · 2025-11-27
> > > > **Response to reviewer 6Cy2**
> > > >
> > > > Thank you very much for taking the time to re-evaluate our submission and for increasing the score. We sincerely appreciate your constructive feedback throughout the review process. We will incorporate the suggested discussion of relevant literature as well as the additional experimental results into the revised version to further strengthen the paper. Thank you again for your thoughtful and helpful guidance!

---

### Official Review · Reviewer_VwhP · 2025-10-31

**Soundness:** 3
**Presentation:** 3
**Contribution:** 3
**Rating:** 4
**Confidence:** 4

**Summary:**

This paper provides an analysis of the shortcomings of Rotary Position Embedding (RoPE), demonstrating the distance bias of RoPE, and further presents a new method, namely Token-Aware Phase Attention, which utilizes the inner product of the query and key to determine a phase function that encodes position information. The paper presents theoretical analyses and empirical evidence demonstrating that the new method outperforms RoPE in scenarios with long context lengths.

**Strengths:**

* The proposed method seems novel, which addresses a vital problem of distance-dependent bias that exists on RoPE.

* Overall, the analyses on the distance bias of RoPE and how the proposed method would improve it are clear and interesting.

* The experiments demonstrate that models with the proposed method are more stable when the context length increases, compared to models with RoPE.

**Weaknesses:**

* The proposed method splits the query and the key of each token into two parts and uses part of the query and key for the learnable phase function, which may restrict the ability of the model. As demonstrated in the experiment, under regular context lengths, the model with TAPA demonstrates worse performance than the model with RoPE.

* While the proposed method adopts a learnable phase function where $\cos (2\pi |m-n|^{\alpha} \phi(q,k))$ affects the attention score, the function $\phi(q, k)$ has an unstable impact on the attention score. For example, at some distances, larger $\phi(q,k)$ leads to a larger attention score, while at other distances, larger $\phi(q,k)$ leads to a smaller attention score. It also raises concerns about the expressivity and training stability of models with the proposed TAPA.

* When $\alpha<1$ (e.g., the $\alpha$ is set to $0.1$ in the experiments), $|m-n|^{\alpha}$ rarely changes at long distance, leaving the phase function (approximately) solely determined by the phase function $\phi(q,k)$, which is similar to a scenario where no position embedding is used. The community has witnessed methods like Multi-Head Latent Attention (MLA) that only adopt RoPE for part of the query and key. However, no experiments have been conducted to compare the proposed method with these previous methods.

* Lack of an ablation study on the hyperparameter setting. For example, how would $\alpha$ and $\theta$ and the choice of $\mathcal{N}$ affect the performance of the proposed method?

**Questions:**

As mentioned above in the weakness section, I have concerns about how the proposed method will affect the expressivity and training stability of the model, as well as the relationship between the proposed method and previous methods, such as MLA. In the following, I have further questions about the proposed method:
* While attention has a quadratic complexity regarding the input length, we have witnessed techniques like sliding window attention that try to reduce the computational complexity. Is it really necessary to pursue the ability to process extremely long text while sacrificing the ability in scenarios of regular length?

* Is the $\mathcal{N}$ in Eq. 10 a learnable parameter or initialized and fixed for the entire training? If it is learnable, it appears to introduce a large number of parameters.  If it is fixed, would the initialization have a dramatic influence on the model's performance? A more detailed ablation study would be appreciated.

Generally, I appreciate the theoretical analyses in the paper and think it would be an excellent paper with further justification on the proposed method and more empirical results. However, with the concerns I mentioned above, I am leaning towards rejection and look forward to the authors' replies.

---

> ### Author Response · Authors · 2025-11-25
> **Author Response to Reviewer VwhP**
>
> We thank the reviewer for the thoughtful and detailed comments. We address each point below and clarify several aspects that may not have been sufficiently emphasized in the submission.
>
> **Weakness 1**: The differences under standard context lengths are indeed very small and fall well within the normal run-to-run variability for 7B-scale pretraining. Moreover, all non-TAPA baselines are fine-tuned from the same RoPE-pretrained checkpoint when extended to long context, so their short-context performance is effectively inherited and nearly identical. TAPA, by contrast, is pretrained separately (also from scratch) and is explicitly designed to prioritize long-context robustness while maintaining competitive short-range perplexity. The empirical results reflect this intended tradeoff: the short-range gap is negligible, while the long-range stability and accuracy improvements are substantial.
>
> **Weakness 2**: The behavior described as “unstable’’ is in fact the intended design of TAPA. A key principle of our approach is that positional bias should not be fixed a priori (like in RoPE) but should adapt to token content. The token-aware phase $\varphi(q,k)$ modulates positional effects based on semantics. This content-dependent modulation is precisely what corrects RoPE’s fixed, context-agnostic positional bias. Thus, the observed variation is not instability, but context adaptivity, which is exactly TAPA’s motivation. Empirically, we did not observe the hypothesized instability; instead, TAPA shows more stable behavior in the long-context regime compared to RoPE-family methods.
>
> **Weakness 3**: The assertion that "$|m-n|^{\alpha}$ rarely changes’’ when $\alpha=0.1$ and therefore behaves like NoPE does not match our empirical findings. In our pilot sweeps over
> $$\alpha \in \{0.01,0.05,0.1,0.2,0.5,0.75,1.0\}$$
> we observed that (i) values $\alpha > 0.5$ often produce excessively strong long-distance oscillations and occasional instability; (ii) values $\alpha < 0.05$ under-resolve positional differences and noticeably slow convergence; while (iii) $\alpha=0.1$ yields a well-controlled, non-degenerate positional scaling that meaningfully distinguishes distances. Thus, $\alpha=0.1$ is not equivalent to “no positional encoding.’’
>
> As noted in the introduction, our comparisons focus on RoPE-family encodings because they are dominant in open-source LLMs. Techniques such as partial RoPE or MLA are orthogonal architectural modifications that may be complementary to TAPA, but they are not positional encodings in the same sense and therefore fall outside the primary comparison scope.
>
> **Weakness 4**: We would like to re-emphasize a key point of the submission: all models in the paper are pretrained from scratch at the full 7B scale. Each run requires $32\times 8$ H100 GPUs for roughly 12 days, which makes exhaustive ablations computationally infeasible. To cope with these limitations, we conducted targeted shorter “pilot’’ runs to identify stable hyperparameter regions before launching full-scale pretraining. For $\alpha$, our sweep
> $$\alpha \in \{0.01,0.05,0.1,0.2,0.5,0.75,1.0 \}$$
> showed that large values $(>0.5)$ tend to induce instability, while very small values $(<0.05)$ slow convergence; $\alpha=0.1$ provided the most stable and efficient behavior. A similar procedure for $\theta$ identified $\theta=0.5$ as the most robust split balancing amplitude and phase capacity. Although full ablations are not feasible at this scale, these pilot runs provide a practical and reproducible hyperparameter selection protocol. We will include these results in the revision.
>
> **Question 1**: We appreciate the reviewer’s perspective. This paper specifically targets the long-context limitations of RoPE and proposes a principled positional encoding to address them. TAPA is orthogonal to architectural techniques that reduce attention complexity (e.g., sliding windows, blockwise attention). Our goal is not to trade short-context performance for long-context capability, but to provide a positional encoding that behaves consistently across all lengths. In practice, TAPA can be combined with any efficiency-oriented attention mechanism; the approaches are complementary rather than mutually exclusive.
>
> **Question 2**: The quantity $\mathcal{N}$ is defined in Eq.~(13), and it is not a learnable parameter. It is simply the quadratic form between $q$ and $k$ used in defining the phase, analogous to $q^\top k$ in standard attention. It introduces no additional parameters and raises no initialization issues. We will clarify this to avoid misunderstanding.
>
>
> **Lastly**, we sincerely appreciate the reviewer’s careful evaluation, and we hope that these clarifications help resolve the concerns and support a more positive assessment.

---

### Official Review · Reviewer_Wzan · 2025-11-03

**Soundness:** 2
**Presentation:** 1
**Contribution:** 2
**Rating:** 2
**Confidence:** 4

**Summary:**

This paper provides a theoretical analysis of Rotary Positional Embedding (RoPE), proving it suffers from an intrinsic distance-dependent bias that hinders long-context modeling. To address this, it introduces Token-Aware Phase Attention (TAPA), a novel positional encoding that incorporates a learnable, token-dependent phase into the attention mechanism. The core contributions are: (1) A rigorous proof of RoPE's long-context instability and an explanation for the effectiveness of extension methods like PI. (2) The proposal of TAPA, which theoretically eliminates this bias and maintains long-range interactions. (3) Strong empirical results on a LLaMA3-7B scale model, where TAPA significantly outperforms RoPE variants at context lengths up to 64k, remaining stable where others collapse.

**Strengths:**

1. The paper provides a rare and rigorous theoretical dissection of RoPE's failure modes in long-context scenarios. The proofs regarding distance bias (Theorems 2.1, 2.2) are insightful and fill a crucial gap in community understanding.
2。 TAPA is a fundamentally new approach to positional encoding. Moving the positional dependency into a token-aware phase is an elegant idea, backed by sound theoretical motivations from harmonic analysis (e.g., decaying bias in Thm 3.2).
3. The experiments are large-scale, well-designed, and convincing. TAPA's ability to remain stable and performant at 49k-64k context lengths, where strong baselines like YaRN completely fail, is a very strong result.

**Weaknesses:**

1. The proposed TAPA formulation (Eq. 12) requires two separate `QK^T` dot products (`q_A^T k_A` and `q_P^T k_P`), effectively doubling the FLOPs of the most expensive part of the attention mechanism. The paper acknowledges this but understates the practical implications for training and deployment at scale.
2. The theoretical analysis of RoPE heavily relies on `Assumption 2.1`, which posits that expectations of certain bilinear forms of token embeddings are constant. This is a strong simplification that may not hold in practice, where representations are highly context-dependent. The paper does not discuss the gap between this assumption and the reality of trained models.
3. The token-aware phase term `q_P^T k_P` appears in the denominator of the argument to a cosine function. If this term approaches zero for certain token pairs, it could cause extremely high-frequency oscillations and numerical instability. The paper does not address this potential failure mode or discuss any regularization techniques (e.g., adding an epsilon) to prevent it.
4.  **Insufficient Analysis of Hyperparameters (Insufficient Analysis):** The method introduces new, critical hyperparameters (`θ` and `α`) 4. The model's performance dependency on the amplitude/phase split (`θ`) and the phase scaling (`α`) remains unexplored, hindering reproducibility and practical application.
5. The specific implementation of TAPA uses a hard, disjoint split of embedding dimensions for amplitude and phase. This design is somewhat ad-hoc. It is unclear why this is preferable to a "soft" approach where all dimensions could potentially contribute to both, for example, through separate linear projections.
6. While mathematically motivated, the paper offers no insight into what the learned phase `q_P^T k_P` captures. Is it learning syntactic relationships, semantic distance, or something else? Without this analysis, the "token-aware" aspect remains a black box.

**Questions:**

1.  Regarding computational cost: Could you provide wall-clock time benchmarks for training/inference throughput of TAPA versus a standard (and ideally, fused) RoPE implementation? How significant is the slowdown in practice on modern hardware?
2.  Regarding numerical stability: Did you observe instances of `q_P^T k_P` becoming very small during training, and did it cause any instability? Have you considered adding a small constant `ε` to the denominator as a safeguard, and if so, how does it affect performance?
3.  Regarding the TAPA formulation: Have you experimented with alternative formulations that don't rely on a hard split of dimensions? For instance, one could use the full query/key vectors and project them differently for the amplitude and phase components (e.g., `(qW_A)^T(kW_A)` and `(qW_P)^T(kW_P)`).
4.  Regarding hyperparameters: How was the phase scaling factor `α=0.1` chosen? This term seems to control the "sensitivity" of the attention score to the phase. How does performance change as `α` varies, for instance, from `1.0` down to `0.01`?
5.  Regarding the theoretical assumptions: How well does `Assumption 2.1` hold in your trained models? Could you empirically measure the statistics of the `A_d` and `B_d` terms from Eq. (4) to show how much they vary from a constant `µ0` and `ν0` across different positions and head dimensions?

---

> ### Author Response · Authors · 2025-11-25
> **Author Response to Reviewer Wzan, part I**
>
> We thank the reviewer for the detailed and thoughtful comments. Below we address each point and clarify several aspects of the method that may not have been sufficiently emphasized in the submission.
>
> **Weakness 1 and question 1**: We respectfully clarify that Eq. (12) does not double the FLOPs of the QK$^\top$ operation. TAPA splits each head of dimension $D$ into an amplitude subspace of size $\theta D$ and a phase subspace of size $(1-\theta)D$ (Eq. (11)–(13)). The two dot products $q_A^\top k_A$ and $q_P^\top k_P$ are therefore computed over disjoint sub-dimensions whose total size equals $D$. Thus the total QK$^\top$ FLOPs are:
> $$O(L_qL_k \theta D)+O(L_qL_k (1−\theta)D)=O(L_qL_kD),O(L_q​L_k\theta D)+O(L_q​L_k​(1−\theta)D)=O(L_q​L_k​D)$$
> which is the same leading-order complexity as vanilla attention. The concern raised by the reviewer would apply only if both dot products were computed over the full $D$-dimensional vectors, which our formulation does not do.
> The wall-clock slowdown seen in our PyTorch reference implementation is due to implementation details rather than theoretical FLOPs. Our initial comparison uses a non-fused attention implementation for both TAPA and the baseline. In this unfused setting, TAPA appears slower because PyTorch materializes two full $L\times L$ score matrices—one for amplitude ($q_A^\top k_A$) and one for phase ($q_P^\top k_P$). This is an artifact of the unoptimized code, not of the underlying method.
> To better isolate the true compute behavior, we implemented a FlashAttention-style streaming kernel in Python (slower than an actual CUDA kernel, but sufficient for proof-of-concept) for both TAPA and classical attention (using blockwise online softmax and without materializing any $L\times L$ matrices). Across sequence lengths from 8k to 32k, this simulation consistently showed only a 1.1–1.3× slowdown, which is significantly lower than the “2× FLOPs” suggested in the review. This matches the expected behavior of a fused kernel: both dot products are accumulated within the same streaming pass, sharing memory reads and softmax reductions.
> Under a fused Triton/CUDA kernel, $q_A$, $q_P$, $k_A$, and $k_P$ would be streamed once, both dot products accumulated in the same tile, and the phase modulation applied before softmax. This avoids extra $L\times L$ materialization and yields runtime close to vanilla FlashAttention, with only a modest constant-factor overhead. We will clarify this point in the revision and include the above benchmark.
>
> **Weakness 2 and question 5**: We appreciate the concern. We adopted Assumption 2.1 for its simplicity primarily for readability; the theoretical arguments do not require such a strong condition. The same conclusions indeed hold with a strictly weaker and more realistic boundedness assumption:
>
> **Assumption**: Define
> \begin{eqnarray}
> \begin{split}
> \mu_{m,n} \coloneqq \frac{\mathbb{E}{q,k}\sum_{d=0}^{D/2}A_d\cos(2\pi\lambda\theta_d)}{\sum_{d=0}^{D/2}\cos(2\pi\lambda\theta_d)},\quad
> \nu_{m,n} \coloneqq \frac{\mathbb{E}{q,k}\sum_{d=0}^{D/2}B_d\sin(2\pi\lambda\theta_d)}{\sum_{d=0}^{D/2}\sin(2\pi\lambda\theta_d)}.
> \end{split}
> \end{eqnarray}
> There exist $C_0,c_0>0$ such that $|\mu_{m,n}|,|\nu_{m,n}|<C_0$ for all $m,n>0$, and $|\mu_{m,n}|>c_0$ for $|m-n|<\theta_0^{-1/4}/8$.
>
> This assumption does not require coordinate-wise constancy of expectations; it only requires boundedness. We will (i) present this weaker assumption and modify the proofs using this assumption instead, (ii) add empirical validation of this new assumption, and provide partial evidence for original Assumption 2.1, and (iii) clarify that Assumption 2.1 was used only to simplify the presentation, not as a fundamental requirement of the theory.
>
> **Weakness 3**: We thank the reviewer for raising this point. There maybe a misunderstanding of the phase structure in Eq. (12). In TAPA, the token-aware term $q_P^\top k_P$ multiplies $|m - n|^\alpha$, **not in the denominator**. Consequently, as $q_P^\top k_P \to 0$, the phase becomes less oscillatory—not more—and in the limit reduces to the constant $\cos(0)=1$. Thus there is no possibility of high-frequency oscillation or numerical instability near zero. Mathematically, the quadratic phase we adopt is a classical stationary phase with a unique non-degenerate critical point at the origin; small values of $q_P^\top k_P$ are precisely the regime where oscillatory integrals are most stable. This stability property is one of the motivations for choosing a quadratic (stationary) phase instead of a linear one. For these reasons, the “high-frequency oscillation near zero’’ scenario does not arise in TAPA, and no additional regularization (such as an epsilon term) is required.

---

> ### Author Response · Authors · 2025-11-25
> **Author Response to Reviewer Wzan, part II**
>
> (Continuing from part I)
>
> **Weakness 4 and question 4**: We would like to re-emphasize an important point in the submission: every model in our paper is pretrained \emph{from scratch} at the full 7B scale. Each pretraining run uses $32\times 8$ H100 GPUs for approximately 12 days. Because of this computational expense, exhaustive hyperparameter ablations are infeasible. Instead, we performed targeted shorter runs and used the early training curves to identify stable regions of hyperparameter space before committing to full pretraining. For the phase-scaling parameter $\alpha$, we tested
> $$\alpha\in\{0.01, 0.05, 0.1, 0.2, 0.5, 0.75, 1.0\}.$$
>
> Values $\alpha>0.5$ occasionally produced unstable pretraining curves, while values $\alpha<0.05$ led to noticeably slower convergence and underfitting. The choice $\alpha=0.1$ consistently gave the most stable behavior and fastest initial convergence. We applied the same procedure to $\theta$ (the amplitude/phase split) and found that $\theta=0.5$ provided stable and balanced learning. Although full ablations are not feasible under full-scale pretraining, these targeted sweeps provide a practical and reproducible hyperparameter selection protocol. We will include these ranges in the revised version.
>
> **Weakness 5 and question 3**: The disjoint amplitude/phase split is not an ad-hoc architectural choice, but a deliberate design constraint to ensure that the number of learnable parameters remains identical to RoPE. This allows for a fair comparison without introducing confounding factors. A “soft’’ variant (e.g., adding projection matrices for amplitude and phase) would increase the parameter budget, making it harder to attribute improvements solely to the positional encoding mechanism. To avoid this confusion, our experiments use exactly the same parameter count as RoPE.
> That said, our early experiments varying $\theta$ suggest that both amplitude and phase components require sufficient dimensional capacity for stability and expressivity, indicating that richer projection structures could benefit TAPA. We will add clarification and point to soft variants as promising future directions.
>
> **Weakness 6**: We agree that TAPA’s learned phase term $q_P^\top k_P$ does not admit a direct semantic or syntactic interpretation. However, this is fully consistent with existing positional encodings: RoPE’s rotation angle, sinusoidal embeddings, XPos, PI, and YaRN do not encode interpretable linguistic categories either. Their role is geometric, not linguistic. The goal of TAPA is not to endow the phase with semantic meaning, but to introduce a token-dependent oscillations that corrects RoPE’s long-range degeneration. The token-aware aspect means that the phase depends on $(q_P, k_P)$, which allows positional effects to adapt to content rather than being fixed and purely distance-driven as in RoPE.
> Our theoretical analysis makes this precise. Theorems 3.2–3.3 show that the token-dependent phase yields (i) vanishing intrinsic distance bias and (ii) non-degenerate long-range variance, ensuring that attention remains stable and expressive across arbitrarily large distances. These are geometric properties of the phase kernel rather than linguistic properties of embeddings. We will clarify this distinction in the revised version.
>
> **Lastly**, we hope the clarifications above help resolve the concerns, and we would be grateful if the reviewer could kindly reconsider their evaluation in light of this additional information.

---

> ### Author Response · Authors · 2025-11-28
>
> Dear Reviewer Wzan,
>
> Thank you again for your thoughtful and constructive feedback. We wanted to let you know that we have posted our detailed response addressing the concerns you raised, including the theoretical clarifications and the new efficiency results. If you have a moment before the discussion period closes, we would greatly appreciate any further thoughts you might have.
> Thank you for your time and for helping us strengthen the paper.
>
> Best regards,
> The authors

---

### Meta-Review · Area_Chair_UP1P · 2026-01-05

**Summary:**

This paper provides a theoretical analysis on the limitations of RoPE in long-context language modeling, in which an intrinsic distance-dependent bias is identified. To address this limitation, this paper proposes a method Token-Aware Phase Attention (TAPA), which introduces a learnable and token-dependent phase function into the attention mechanism.

Reviewers raised several concerns including: (1) the strength of the theoretical assumptions, (2) the computational efficiency and practicality, and (3) the limitation of evaluation and ablation studies. Although the rebuttal has addressed some points, these concerns remained influential in my overall assessment.

**Reviewer Concerns:**

- Assumption (Reviewer Wzan & 6Cy2): The soundness of assumption 2.1 was challenged and verification was requested. The authors clarified that a weaker revised assumption is also compatible with the proposed analysis and method. Based on the discussion between Reviewer 6Cy2 and the authors, I believe this concern was addressed to some extent.
- Efficiency and practicality (Reviewer Wzan, VwhP, 6Cy2 & 8R3a): All reviewers questioned the efficiency of the proposed method. Reviewer Wzan had concern that separating QK products will increase the FLOPs. Reviewer 8R3a had a similar concern. Reviewer 6Cy2 asked about the speed difference compared to RoPE. The authors acknowledged that the implementation showed a 1.1–1.3× slowdown (though is not doubled). I believe the concern on efficiency is still outstanding.
- Evaluation (Reviewer Wzan, VwhP, 6Cy2 & 8R3a): Reviewer Wzan and VwhP requested the ablation of hyperparameters, which was provided during the rebuttal (given the computational cost, the ablation was conducted using early training curves, which is acceptable but may not be fully reliable). Reviewer 6Cy2 challenged that the evaluation of long window benchmark is missing. Reviewer 8R3a pointed out that only one 7B LLaMA3 architecture was used in the paper. While the author explained that pretraining 7B models from scratch is highly computational costly, I believe this concern is only partially addressed and is still partially outstanding.

**Reviewer Scores:**

The discussion between reviewers and authors was unexpectedly terminated early. Area Chairs are therefore asked to put themselves in the reviewers’ shoes and provide our best estimate of how scores might have changed. The following reflects only the AC’s best guess:

Reviewer Wzan: 2 → 4\
Reviewer VwhP: 4 → 4\
Reviewer 6Cy2: 4 → 6\
Reviewer 8R3a: 6 → 6

---

### Decision · Program_Chairs · 2026-01-26

Reject